# Random Neural Network Expressivity for Non-Linear Partial Differential Equations

## Abstract

Neural networks with randomly generated hidden weights (RaNNs) have been extensively studied, both as a standalone learning method and as an initialization for fully trainable deep learning methods. In this work, we study RaNN expressivity for learning solutions to non-linear partial differential equations (PDEs). To achieve this, we derive approximation error bounds for time-dependent Sobolev functions and obtain a dimension-free approximation rate $\frac{1}{2}$. Our results imply that RaNNs are capable of efficiently approximating solutions to complex non-linear PDEs. When applied to Physics-Informed Neural Networks (PINNs), our bounds imply that with high probability, the physics-informed training error converges to $0$ with convergence rate free from the curse of dimensionality. Our theoretical analysis is supported by numerical experiments on two benchmark PDEs. These simulations validate the obtained convergence rate.

## 1 Introduction

Partial Differential Equations (PDEs) are foundational to our understanding of the natural world, with applications across all areas of science and engineering. Many complex phenomena are modelled by non-linear PDEs (e.g. Navier-Stokes, Schrödinger, Porous medium equations), which exhibit disorderly behaviour that renders them intractable to classical analytic/numerical approaches. Therefore, it is crucial to develop numerical methods for solving non-linear PDEs efficiently. In the past years, a variety of deep learning methods for solving PDEs have been introduced and analysed. Neural networks with randomly generated hidden weights (RaNNs) play an important role in many of these methods; either as standalone learning method or as initialization for fully trainable deep neural networks. In both cases, when employing these methods, a precise understanding of the approximation error is crucial for controlling the overall error.

In this paper, we are concerned with expressivity of RaNNs for learning solutions of non-linear PDEs. To tackle this problem, we derive approximation error bounds for time-dependent Sobolev functions, which encompass the solution spaces for many important non-linear PDEs. Our obtained bounds show that these functions can be approximated by RaNNs at rate $\frac{1}{2}$, making them efficient also in high-dimensional situations. We then apply our results to two important non-linear PDEs and show that with high probability the training error for Physics-Informed Neural Networks (PINNs) (Raissi et al., 2019) converges to $0$ with rate free from the curse of dimensionality. We complement our theoretical analysis by numerical experiments on two benchmark PDEs, validating the obtained convergence rate. In particular, our results provide quantitative approximation guarantees for RaNN-based PINNs for learning non-linear PDEs, as have been extensively studied in computational experiments (cf. the references below).

### 1.1 Related Works

In recent years, a variety of deep learning-based methods for solving PDEs have been introduced, addressing challenges of classical mesh-based methods such as finite difference methods. Seminal works include Sirignano & Spiliopoulos (2017), E et al. (2017) Raissi et al. (2019), E & Yu (2017). We refer, e.g., to the survey articles Beck et al. (2020); Germain et al. (2021); Cuomo et al. (2022); Gonon et al. (2024) for an extensive overview and further references on deep learning methods for PDEs and their theoretical foundations.

PINNs (Raissi et al., 2019) constitute a flexible and widely applicable deep learning-based approach for solving PDEs. PINNs reframe the problem as training a neural network to solve the PDE, by minimising a loss function that encodes the PDE residual along with the boundary/initial conditions. While this approach has been demonstrated to be highly effective in many settings Cai et al. (2021); Hu et al. (2024), for non-linear PDEs the loss landscape may become exceptionally complex. This has motivated the use of RaNN-based PINN methods, for which several recent studies have carried out extensive empirical experiments, see, e.g., Dwivedi & Srinivasan (2020); Shang et al. (2023); Shang & Wang (2024); Sun et al. (2024); Wang & Dong (2024); Ying et al. (2024); Linghu et al. (2025); Datar et al. (2025); Chen et al. (2022); Nelsen & Stuart (2021).

RaNNs Huang et al. (2006); Rahimi & Recht (2007; 2008) are neural networks with randomly generated hidden weights. RaNNs have been used both as standalone learning methods and as means for studying the effects of random initialization for neural networks trained using gradient-based optimization Braun et al. (2024); Carratino et al. (2018). Generalization properties of random feature models have been studied in Rudi & Rosasco (2017); Mei & Montanari (2019); Lanthaler & Nelsen (2023); Cheng et al. (2023). In addition to RaNN-based PINNs, many other RaNNs-based methods have been developed for solving PDEs Nelsen & Stuart (2021); Gonon (2023); Jacquier & Zuric (2023); Neufeld et al. (2025). More broadly, RaNNs and related random feature models have demonstrated state-of-the-art performance and speed across various tasks Bolager et al. (2023); Dempster et al. (2023); Gattiglio et al. (2024); Prabhu et al. (2024); Zozoulenko et al. (2025).

The reduced number of trainable parameters of RaNNs in comparison to fully trainable models results in a simpler training phase with reduced computational cost, potentially at the expense of lower expressivity. Therefore, a precise theoretical understanding of RaNN approximation capabilities is crucial. Quantitative approximation properties of RaNNs for functions in the associated reproducing kernel Hilbert space have been studied in Rahimi & Recht (2008); Bach (2017); Sun et al. (2018). For smoothness-based function classes, RaNN approximation error bounds were derived in Gonon et al. (2023); Gonon (2023) using Barron-type representations and further extended in Neufeld & Schmocker (2023); De Ryck et al. (2025). In the context of PDEs, Gonon (2023) obtains a full RaNN learning error analysis free from the curse of dimensionality for a class of linear PDEs. In all these results, the random weight distribution is fixed (e.g. a uniform, normal or Student-t distribution). In contrast, RaNNs also appear as a means of proof for deriving deterministic approximation bounds Barron (1993; 1994); Barron & Klusowski (2018); Siegel & Xu (2020), with RaNN weight distributions depending on the function to be approximated.

While the approximation results in Gonon et al. (2023); Gonon (2023) and Neufeld & Schmocker (2023); De Ryck et al. (2025) allow to control the RaNN approximation errors in uniform, mean-squared or Sobolev-norms, respectively, in the context of non-linear PDEs these results would either require strict information on the solution (e.g. finiteness in Barron-ridgelet norms and decay on the Fourier transform of $u$) which is typically not known, or the bounds would be applicable only for approximating PDE solutions at a fixed point in time. However, for time-dependent non-linear PDEs, solutions often have significantly different behaviour in time versus space (e.g. solutions to Navier-Stokes or semi-linear heat equations). In contrast, our results allow to handle time-dependent functions in mixed Sobolev spaces, as arise in the context of non-linear PDEs.

## 1.2 CONTRIBUTIONS

In this paper, we provide RaNN approximation error bounds tailored to the context of time-dependent, non-linear PDEs. We will denote the width of a RaNN by $N$. Our paper makes the following contributions:

1. **RaNN approximation bound with dimension-independent rate** $N^{-1/2}$**:** We derive RaNN approximation error bounds for time-dependent Sobolev functions (Theorem 1). These functions encompass the solution spaces for many important non-linear PDEs. Our unbiased RaNN estimator approximates functions in mixed Sobolev norms $H_t^p H_x^q$ at the rate $N^{-1/2}$ independently of dimension, while only requiring minimal extra regularity in time and space.

2. **Implications for non-linear PDEs:** We showcase the implications of our bounds on two important classes of non-linear PDEs: Porous Medium Equations (PME) and Compressible Navier-Stokes Equations. In particular, we obtain RaNN approximation error bounds

for the PINN training error in these cases. Our results are supplemented by numerical simulations validating the obtained convergence rates.

To prove these results, we obtain a specific ridgelet-based representation for $L^2$ functions (Proposition 1) and a higher-order Plancherel-type estimate (Lemma 1) that connects Sobolev regularity of $u$ with its ridgelet transform. Our obtained bounds may serve as building block for generalization error analyses of RaNNs for PINN-based learning as obtained for deterministic networks in De Ryck & Mishra (2024); Mishra & Molinaro (2022; 2023); Alejo et al. (2024).

## 2 PRELIMINARIES

### 2.1 RANDOM NEURAL NETWORKS

A random neural network (RaNN) is a fully connected neural network with one hidden layer in which the weights are randomly sampled, leaving only the output weights trainable. In this paper, we are interested in studying solutions to PDEs, which typically treat time as a separate dimension. Therefore we will consider time-dependent random neural networks of the following form.

**Definition 1.** *A time-dependent random neural network of width $N$ is a function $u_W^{\tau,\mathbf{a},\mathbf{b}} : \mathbb{R} \times \mathbb{R}^d \to \mathbb{R}$ with*

$$u_W^{\tau,\mathbf{a},\mathbf{b}}(t,x) = \sum_{i=1}^{N} W_i \sigma(\tau_i t + \mathbf{a}_i \cdot x + b_i), \tag{1}$$

*where $\sigma : \mathbb{R} \to \mathbb{R}$ is an activation function, $\mathbf{a}_1, ..., \mathbf{a}_N$ are $\mathbb{R}^d$-valued i.i.d. random variables, $b_1, ..., b_N$ are i.i.d. random variables in $\mathbb{R}$, $\tau_1, ..., \tau_N$ are i.i.d. random variables in $\mathbb{R}$ and $W_1, ..., W_N$ are trainable weights.[1]*

A RaNN is used as learning system by optimizing the weights $W_1, ..., W_N$ of $u_W^{\tau,\mathbf{a},\mathbf{b}}$ with respect to a given loss function.

### 2.2 PHYSICS-INFORMED MACHINE LEARNING

Physics-informed neural networks (PINNs) have been introduced in Raissi et al. (2019) as an unsupervised learning method for solving partial differential equations. PINNs approximate the solution $u$ to a PDE $\mathcal{L}[u] = 0$ by a neural network $u_\theta$, with $\theta$ representing the trainable parameters. The solution is approximated by minimising a loss function $\mathcal{J}[u_\theta]$ encoding the structure of the PDE — including initial/boundary conditions — at collocation points $\{t_p^i, x_p^i\}_{i=1}^{M}$ in the interior domain $(0,T) \times D$, as well as $\{x_{ic}^i\}_{i=1}^{M}$, $\{t_{bc}^i\}_{i=1}^{M}$ on the slices $\{t = 0\} \times D$ and $(0,T) \times \partial D$ on which the initial data / boundary conditions are defined, respectively. For example, if we have a PDE on the 1D domain $(0,T) \times (a,b)$ given by $\mathcal{L}[u] = 0$ with initial condition $u(0,\cdot) = u_0$ and boundary conditions $u(t,a) = u(t,b)$, the PINN loss function is given by

$$\mathcal{J}[u_\theta] := \frac{1}{M} \sum_{i=1}^{M} |\mathcal{L}[u_\theta]|^2(t_p^i, x_p^i) + \frac{1}{M} \sum_{i=1}^{M} |u_\theta(0, x_{ic}^i) - u_0(x_{ic}^i)|^2 + \frac{1}{M} \sum_{i=1}^{M} |u_\theta(t_{bc}^i, b) - u_\theta(t_{bc}^i, a)|^2.$$

The parameters $\theta$ are then iteratively updated using a stochastic optimization algorithm.

### 2.3 RIDGELET TRANSFORM

The ridgelet transform $\mathcal{R}_\psi u$ of $u : \mathbb{R} \times \mathbb{R}^d \to \mathbb{R}$ with respect to $\psi : \mathbb{R} \to \mathbb{R}$ is given by

$$\mathcal{R}_\psi u(\tau, \mathbf{a}, b) := \int_{\mathbb{R}^{d+1}} u(t, \mathbf{x}) \psi(\tau t + \mathbf{a} \cdot \mathbf{x} - b) \|(\tau, \mathbf{a})\|^s \, dt d\mathbf{x}, \qquad \tau \in \mathbb{R}, \mathbf{x} \in \mathbb{R}^d, b \in \mathbb{R}. \tag{2}$$

The factor $\|(\tau, \mathbf{a})\|^s$ appears for convenience in the literature. We will take $s = 0$ in this paper. Then the dual ridgelet transform $R_\eta^\dagger T$ of $T : \mathbb{R}^{d+2} \to \mathbb{R}$ with respect to $\eta : \mathbb{R} \to \mathbb{R}$ is defined as

$$\mathcal{R}_\eta^\dagger T(t, \mathbf{x}) := \int_{\mathbb{R}^{d+2}} T(\tau, \mathbf{a}, b) \eta(\tau t + \mathbf{a} \cdot \mathbf{x} - b) \|(\tau, \mathbf{a})\|^{-s} d\tau d\mathbf{a} db. \tag{3}$$

---

[1] Formally, $W_i = g_i(\mathbf{a}_1, ..., \mathbf{a}_N, b_1, \ldots, b_N)$ for measurable functions $g_i$.

We refer to Sonoda & Murata (2017); Murata (1996) for a more comprehensive overview of the ridgelet transform and its properties.

## 2.4 Notation and structure of the paper

We adopt the shorthand notation $X_t Y_x$ for the Bochner space $X(\mathbb{R}; Y(\mathbb{R}^d))$ (or $X(0, T; Y(D))$, depending on the context). For example, we may write $H_t^p H_x^q$ for the space $H^p(\mathbb{R}; H^q(\mathbb{R}^d))$. We write $\widehat{f}$ to denote the Fourier transform of $f$, with the convention $\widehat{f}(\omega) = \int e^{-i\omega x} f(x)\, dx$. We also denote by $\|\cdot\|$ the $\ell^2$ norm. In Section 3, we prove a representation formula for general time-dependent Sobolev functions, before proving a key inequality that connects the regularity of the ridgelet transform to the original function. We use this to derive our main result Theorem 1. In Section 4 we then apply Theorem 1 to RaNN-PINN approximators of two benchmark PDEs.

## 3 RaNN approximations of time-dependent Sobolev functions

Our first step is to obtain an integral representation for $u$ based on the ridgelet transform. This representation will be used later to derive RaNN approximation error bounds for $u$.

### 3.1 Obtaining an integral representation

We first introduce the Lizorkin distribution space $\mathcal{S}_0'(\mathbb{R})$, which is the dual space of $\mathcal{S}_0(\mathbb{R})$; the space of Lizorkin functions. $\mathcal{S}_0(\mathbb{R})$ is a closed subspace of the space of Schwartz functions $\mathcal{S}(\mathbb{R})$ and contains the functions $f \in \mathcal{S}(\mathbb{R})$ such that all moments vanish, i.e. $\mathcal{S}_0(\mathbb{R}^d) = \{f \in \mathcal{S} : \int_{\mathbb{R}^d} \mathbf{x}^\alpha f(\mathbf{x})\, d\mathbf{x} = 0$ for any $\alpha \in \mathbb{N}_0^d\}$. The Lizorkin distribution space $\mathcal{S}_0'(\mathbb{R})$ itself includes many common activations such as $\tanh$, sigmoid and ReLU. We refer to Sonoda & Murata (2017) for a more detailed description of the $\mathcal{S}_0'(\mathbb{R})$ space. In this paper, we will consider activation functions $\sigma : \mathbb{R} \to \mathbb{R}$ that belong to the following subspace of $\mathcal{S}_0'(\mathbb{R})$ for some $k \geq 0$.

**Definition 2.** *Let $k \in \mathbb{N}_0$. We say $\sigma \in \mathcal{S}_0'(\mathbb{R})$ belongs to $\mathcal{T}_k$ if (i) there exists $C_1 > 0$ such that*

$$\sum_{j=0}^{k} |\sigma^{(j)}(x)| \leq C_1 \qquad \forall x \in \mathbb{R}, \tag{4}$$

*and (ii) there exists $\delta > 0$ and $\beta \in \mathbb{N}_0$ such that $\zeta^\beta \widehat{\sigma}(\zeta) \in C(-\delta, \delta)$ and for any $\alpha \in \mathbb{N}_0$*

$$J_\sigma := \int_{\mathbb{R}} \zeta^{2\alpha+\beta} \widehat{\sigma}(\zeta) e^{-\zeta^2/2}\, d\zeta \neq 0. \tag{5}$$

Examples of admissible activation functions for which our results hold are $\tanh, \cos$ and sigmoid (each in $\mathcal{T}_k$ for all $k \geq 0$). This is shown in Remark 3 in the appendix. For $\mathcal{S}(\mathbb{R})$ and $m \in \mathbb{N}$ we also define the following (possibly infinite) admissibility constant

$$A_{\psi,m} := \int_{-1}^{1} |\widehat{\psi}(\omega)|^2 |\omega|^{-m}\, d\omega. \tag{6}$$

We now mention the following integral representation formula, which directly follows from results obtained by Sonoda & Murata (2017) using the theory of ridgelet transforms.

**Proposition 1.** *Let $m \geq 0$ and $\sigma \in \mathcal{T}_k$ for some $k \geq 0$. A function $u \in L^2(\mathbb{R}; L^2(\mathbb{R}^d))$ can be expressed as*

$$u(t, \mathbf{x}) = \int_{\mathbb{R}} \int_{\mathbb{R}^d} \int_{\mathbb{R}} (\mathcal{R}_\psi u)(\tau, \mathbf{a}, b) \sigma(\tau t + \mathbf{a} \cdot \mathbf{x} - b)\, d\tau d\mathbf{a} db, \tag{7}$$

*where $\mathcal{R}_\psi u$ is the ridgelet transform of $u$ with respect to a Schwartz function $\psi \in \mathcal{S}(\mathbb{R})$ such that*

$$|\widehat{\psi}(\omega)| \leq C|\omega|^m \qquad \forall\, |\omega| < 1, \tag{8}$$

*for some $C > 0$ independent of $u$, and therefore $A_{\psi,m} < +\infty$. We also have that $(\sigma, \psi)$ is an admissible pair in the sense of Sonoda & Murata (2017), meaning that the following constant is finite and non-zero:*

$$K_{\psi,\sigma} := (2\pi)^{d-1} \int_{\mathbb{R}} \frac{\hat{\psi}(\zeta)\hat{\sigma}(\zeta)}{|\zeta|^m}\, d\zeta. \tag{9}$$

The construction of an appropriate $\psi$ is given in the proof of Proposition 1 in Appendix A.1. From now on, for any activation $\sigma \in \mathcal{T}_k$ and function $u \in L^2(\mathbb{R}; L^2(\mathbb{R}^d))$ we will denote by $\mathcal{R}_\psi u$ the ridgelet transform of $u$ with respect to $\psi \in \mathcal{S}(\mathbb{R})$ constructed according to Proposition 1.

## 3.2 PARSEVAL RELATION FOR THE RIDGELET TRANSFORM

In order to derive RaNN approximation error bounds, we will need a Parseval-type result which connects the regularity of $\mathcal{R}_\psi u$ in parameter space with the regularity of $u$ in cartesian space. For convenience, we give an outline of the proof in Appendix A.2 and the full proof in Appendix A.3.

**Lemma 1.** *Suppose $u \in H^p(\mathbb{R}; H^q(\mathbb{R}^d))$ for some $p, q \geq 0$, and that $u$ is compactly supported in time and space on $[-2T, 2T] \times [-2R, 2R]^d$ for some $T, R \geq 0$. Then there exists $\psi \in \mathcal{S}(\mathbb{R})$ with*

$$I := \int_{\mathbb{R}^{d+2}} |\mathcal{R}_\psi u(\tau, \mathbf{a}, b)|^2 (1 + |\tau|^2)^p (1 + \|\mathbf{a}\|^2)^q (1 + b^2) \, d\tau d\mathbf{a} db \tag{10}$$

$$\leq \mathcal{L}_\psi \|u\|^2_{H^{p+1}(\mathbb{R}; H^{q+1}(\mathbb{R}^d))},$$

*where, for $M = (2p + 2q + d + 3)/2$,*

$$\mathcal{L}_\psi := (4\pi + \|\psi\|_{L^2(\mathbb{R})})(1 + 4T + 4R) + 4\pi(M + 1)^2 + \|\psi'\|^2_{L^2(\mathbb{R})}. \tag{11}$$

## 3.3 RANDOM NEURAL NETWORK APPROXIMATION ERROR BOUNDS

In this subsection we provide our main result for approximating a function $u \in H^p(0, T; H^q(D))$ using RaNNs. The proof is given as an outline, and the full details are deferred to Appendix A.4.

**Theorem 1.** *Fix a bounded subset $(0, T) \times D \subset \mathbb{R}_+ \times \mathbb{R}^d$ and let $u \in H^{p+s_1}(0, T; H^{q+s_2}(D))$ for $p, q \geq 0, s_1 > 3/2$ and $s_2 > (d + 2)/2$. Furthermore, let $\sigma \in \mathcal{T}_{p+q}$. There exist weights $\{W_i\}_{i=1}^N$ such that the following random neural network is an unbiased estimator of $u$:*

$$u_N(t, \mathbf{x}) = \sum_{i=1}^N W_i \sigma(\tau_i t + \mathbf{a}_i \mathbf{x} + b_i)$$

*for $(\tau_i, \mathbf{a}_i, b_i) \sim \pi$, where*

$$\pi(t, \mathbf{a}, b) = \frac{1}{C_\pi}(1 + \tau^2)^{-\lambda_\tau}(1 + \|\mathbf{a}\|^2)^{-\lambda_a}(1 + b^2)^{-1}, \quad \lambda_\tau > 1/2, \lambda_a > d/2, \tag{12}$$

*and $C_\pi$ is the normalisation constant*

$$C_\pi := \int_{\mathbb{R}} \frac{1}{(1 + \tau^2)^{\lambda_\tau}} \, d\tau \cdot \int_{\mathbb{R}^d} \frac{1}{(1 + \|\mathbf{a}\|^2)^{\lambda_a}} d\mathbf{a} \cdot \int_{\mathbb{R}} \frac{1}{(1 + b^2)} \, db. \tag{13}$$

*The neural network $u_N$ satisfies*

$$\mathbb{E}_\Theta \left( \|u - u_N\|^2_{H^p(0,T;H^q(D))} \right) \leq \frac{C_\Omega C_\pi \|\sigma^{(p+q)}\|^2_\infty T |D| (p+q) \mathcal{L}_\psi}{N} \cdot \|u\|^2_{H^{p+s_1}(\mathbb{R}; H^{q+s_2}(\mathbb{R}^d))}, \tag{14}$$

*where $C_\Omega$ is a constant dependent on $p, q, d$ and the domain, and $\mathcal{L}_\psi$ is given by (11).*

**Remark 1.** *This is an improvement upon the work of De Ryck et al. (2025) (see Theorem 3.9) which required $u \in H^s(\mathbb{R}^d)$ for $s \geq (d + 9)/2$ for estimates in $H^1(\mathbb{R}^d)$ and $H^2(\mathbb{R}^d)$, with slower rates. Moreover, we obtain estimates in a time-dependent norm, whereas previous results only obtained error rates for solutions at a fixed time (see also Proposition 4.24 of Neufeld & Schmocker (2023)).*

*Proof (Outline).* **Step 1: extension of $u$ to $\mathbb{R} \times \mathbb{R}^d$.** We wish to use Proposition 1 and Lemma 1, which each assume that $u$ is defined on $\mathbb{R} \times \mathbb{R}^d$. Lemma 1 in particular also assumes compact support in time and space. To adhere to these constraints, we construct $\tilde{u}$ to be a smooth extension of $u$ which satisfies $u = \tilde{u}$ on $(0, T) \times D$, is norm preserving ($\|\tilde{u}\|_{H_t^p H_x^q} \leq C_\Omega \|u\|_{H_t^p H_x^q}$) and compactly supported on $[-2T, 2T] \times [-2R, 2R]^d$. Such an extension is known to exist (cf. Chap. VI of Stein (1970)). The extension constant $C_\Omega$ will generally depend on $p, q, d$ and the domain. For the simple case $p = q = 0$, a zero extension guarantees $C_\Omega = 1$ independently of dimension.

**Step 2: construction of the unbiased estimator $u_N$.** We define $u_N$ as

$$u_N(t, \mathbf{x}) := \frac{1}{N} \sum_{i=1}^{N} \frac{R_\psi \tilde{u}(\tau_i, \mathbf{a}_i, b_i)}{\pi(\tau_i, \mathbf{a}_i, b_i)} \sigma(\tau_i t + \mathbf{a}_i \cdot \mathbf{x} - b_i) \equiv \frac{1}{N} \sum_{i=1}^{N} X_i(t, \mathbf{x}). \quad (15)$$

so that $\mathbb{E}_\Theta(u_N) = \tilde{u}$, where $\Theta = \{(\tau, \mathbf{a}, b) \in \mathbb{R} \times \mathbb{R}^d \times \mathbb{R}\}$ is the parameter space. More generally, for any $0 \leq \ell \leq p$ and multi-index $\beta$ with $|\beta| \leq q$, we have $\mathbb{E}(\partial_t^\ell D_\mathbf{x}^\beta(u_N)) = \partial_t^\ell D_\mathbf{x}^\beta(\tilde{u})$. We can use this to obtain the equality (since $u = \tilde{u}$ on $(0, T) \times D$)

$$\mathbb{E}_\Theta \left( \|\partial_t^\ell D_\mathbf{x}^\beta(u - u_N)\|_{L^2((0,T) \times D)}^2 \right) = \mathbb{E}_\Theta \left( \|\partial_t^\ell D_\mathbf{x}^\beta(\tilde{u} - u_N)\|_{L^2((0,T) \times D)}^2 \right)$$

$$= \frac{1}{N} \int_{(0,T) \times D} \mathbb{E}_\Theta |\partial_t^\ell D_\mathbf{x}^\beta X_i|^2 \, dxdt.$$

**Step 3: bounding $\mathbb{E}_\Theta(|\partial_t^\ell D_\mathbf{x}^\alpha X_i|^2)$.** To proceed from the above equation, we compute $\mathbb{E}_\Theta(|\partial_t^\ell D_\mathbf{x}^\alpha X_i|^2)$ from (15). This gives us

$$\mathbb{E}_\Theta(|\partial_t^\ell D_\mathbf{x}^\alpha X_i(t, \mathbf{x})|^2)$$

$$\leq C_\pi \int_{\mathbb{R}^{d+2}} |R_\psi \tilde{u}(\tau, \mathbf{a}, b)|^2 |\sigma^{(p+q)}|^2 |\tau|^{2p} \|\mathbf{a}\|^{2q} (1 + \tau^2)^{\lambda_\tau} (1 + \|\mathbf{a}\|^2)^{\lambda_a} (1 + b^2) \, d\tau d\mathbf{a} db. \quad (16)$$

Using $|\tau|^{2p} \leq (1 + \tau^2)^p$ and $\|\mathbf{a}\|^{2q} \leq (1 + \|\mathbf{a}\|^2)^q$, we can invoke the inequality (10) and $\sigma \in \mathcal{T}_{p+q}$ to get the result (14). Here, we use the embedding inequality $\|\tilde{u}\|_{H_t^p H_x^q} \leq C_\Omega \|u\|_{H_t^p H_x^q}$. The full details are deferred to the appendix. $\qquad \square$

## 4 APPLICATIONS TO NON-LINEAR PDEs

We look at two representative non-linear PDEs (the Porous Medium Equation and compressible Navier-Stokes) in order to understand how one can obtain specific asymptotic bounds on the residual loss and approximation error using the structure of the PDE.

### 4.1 POROUS MEDIUM EQUATION

The porous medium equation (PME) is an important example of a non-linear parabolic PDE that models the flow of gases through porous mediums. In dimension $d \in \mathbb{N}$ and for $m > 0$, a function $u : \mathbb{R}_+ \times \mathbb{R}^d \to \mathbb{R}$ is said to solve the porous medium equation if

$$\begin{cases} \partial_t u - \Delta(u^m) = 0, \text{ on } \mathbb{R}_+ \times \mathbb{R}^d. \\ u(0, \cdot) = u_0, \end{cases} \quad (17)$$

In the case where $u_0$ is positive and in $H^k(\mathbb{R}^d)$ for some $k \in \mathbb{N}$, the following result is classically known.

- Vázquez (2007): Consider initial data $u_0$ satisfying

$$u_0 \in H^k(\mathbb{R}^d), \ k \in \mathbb{N}, \ 0 < c \leq u_0 \leq C, \quad (18)$$

Then there exists a classical solution $u$ to (17) with $c \leq u(t, x) \leq C$ and

$$u \in C^\infty((0, \infty) \times \mathbb{R}^d) \cap C([0, \infty); H^k(\mathbb{R}^d)).$$

In practice, we simulate solutions on a bounded domain $(0, T) \times D$. In this case, the smoothness of the solution in fact implies that $u \in H^k((0, T) \times D)$ for any $k \geq 0$. We can deduce an approximation result on solutions to PME using Theorem 1. Before we state the result, let us note that the classical loss function one would use when training a physics-informed neural network to approximate solution $u$ is a discretisation of the following metric:

$$\mathcal{J}_{PDE}(u_N) = \int_{(0,T) \times D} |\partial_t u_N - \Delta(u_N^m)|^2 \, dtdx. \quad (19)$$

**Corollary 1.** *Suppose $u$ solves (17) with initial data $u_0 \in H^1(\mathbb{R}^d)$ satisfying (18). Then there exists a random neural network $u_N(t, x)$ such that on the domain $(0, T) \times D$:*

1. *For any $p, q \geq 0$,*

$$\mathbb{E}_\Theta(\|u - u_N\|^2_{H_t^p H_x^q}) \leq \frac{C_\Omega C_\pi \|\sigma^{(p+q)}\|^2_\infty T|D|(p+q)\mathcal{L}_\psi}{N} \|u\|^2_{H_t^{p+s_1} H_x^{q+s_2}} \tag{20}$$

$$=: \frac{\mathcal{M}_\psi}{N} \|u\|^2_{H_t^{p+s_1} H_x^{q+s_2}},$$

   *for any $s_1 > 3/2$, $s_2 > (d+2)/2$.*

2. *For any $\delta \in (0, 1)$, with probability $1 - \delta$ over the network parameters, the PINN training loss can be bounded as:*

$$\mathcal{J}_{PDE}(u_N) \leq \frac{C_m(L + C_{emb}\|u\|^2_{L_t^\infty H_x^{2+k}})\mathcal{M}_\psi}{N\delta} \|u\|^2_{H_t^2 H_x^{3+k}}, \tag{21}$$

   *for $k > d/2$, if the sampled network parameters are such that $u_N$ satisfies $\|u_N\|_{L_{t,x}^\infty} + \|\nabla u_N\|_{L_{t,x}^\infty} + \|\Delta u_N\|_{L_{t,x}^\infty} \leq L < +\infty$. Here, $C_{emb}$ is the constant arising from the Sobolev embedding $H_x^k \hookrightarrow L_x^\infty$ and $C_m$ is a polynomial in the PME parameter $m$.*

*In other words, one can find a sequence of neural networks $u_N$ which drive the PDE residual to $0$.*

### 4.2 Compressible Navier-Stokes Equations

We now look at a more delicate example, which is a system of equations that does not enjoy the instantaneous regularisation property of the PME. The compressible Navier-Stokes equations in dimension $d$ are given by

$$\begin{cases} \partial_t \rho + \text{div}(\rho \mathbf{u}) = 0, & \text{on } (0, T) \times D. \\ \partial_t(\rho \mathbf{u}) + \text{div}(\rho \mathbf{u} \otimes \mathbf{u}) - \nabla(\mu(\rho)\text{div } \mathbf{u}) + \nabla p(\rho) = 0, & \text{on } (0, T) \times D, \end{cases} \tag{22}$$

where

$$p(\rho) = \rho^\gamma, \ \gamma > 0 \text{ and } \mu(\rho) = \rho^\alpha, \ \alpha > 0. \tag{23}$$

The solution is a pair $(\rho, \mathbf{u})$ where the density $\rho : \mathbb{R}_+ \times \mathbb{R}^d \to \mathbb{R}$ is the scalar density and $\mathbf{u} : \mathbb{R}_+ \times \mathbb{R}^d \to \mathbb{R}^d$ is the vector-valued velocity. We look at the one-dimensional setting $D = (0, 1)$ with periodic boundary conditions, where global-in-time classical solutions exist under mild assumptions. In this case, the following global well-posedness result applies.

- Theorem 1.5, Constantin et al. (2020) Given initial data $(\rho_0, u_0)$ satisfying

$$(\rho_0, u_0) \in H^k(\mathbb{R}), \ k \geq 3, \ 0 < \delta \leq \rho_0 \leq C, \tag{24}$$

   then there exists a unique solution $(\rho, u)$ on $(0, T)$ to (22) with initial data $(\rho_0, u_0)$ such that

$$\rho \in C(0, T; H^k(D)), \ u \in C(0, T; H^k(D)) \cap L^2(0, T; H^{k+1}(D)). \tag{25}$$

We will take the pressureless case $p = 0$ and constant viscosity $\mu(\rho) = \mu$ for simplicity, although our computations can be easily extended to handle a more general setting where $p, \mu$ are smooth and convex (e.g. $p(\rho) = \rho^\gamma, \mu(\rho) = \rho^\alpha$, for $\alpha, \gamma > 0$). In this case, the PINN residual loss is the discretisation of the following loss functions:

$$\mathcal{J}^1_{PDE}(\mathbf{v}_N) := \int_{(0,T) \times D} |\partial_t(\rho_N) + \partial_x(\rho_N u_N)|^2 \, dxdt,$$

$$\mathcal{J}^2_{PDE}(\mathbf{v}_N) := \int_{(0,T) \times D} |\partial_t(\rho_N u_N) + \partial_x(\rho_N u_N^2) - \mu\partial_x^2 u_N|^2 \, dxdt. \tag{26}$$

To apply Theorem 1, let's first define the Sobolev product norm $\|(f, g)\|^2_{H_t^p H_x^q} := \|f\|^2_{H_t^p H_x^q} + \|g\|^2_{H_t^p H_x^q}$. We then have the following result whose proof is given in Appendix A.6.

**Corollary 2.** *Suppose $(\rho, u)$ is a solution to (22) in dimension $d = 1$ generated by initial data satisfying $(\rho_0, u_0) \in H^k(\mathbb{R})$, $k \geq 5$, $0 < \delta \leq \rho_0 \leq C$. Then there exists a random neural network $\mathbf{v}_N = (\rho_N, u_N)$ such that*

- *For any $p, q \geq 0$,*

$$\mathbb{E}_\Theta(\|(\rho, u) - \mathbf{v}_N\|_{L_t^2 H_x^q}^2) \leq \frac{\mathcal{M}_\psi}{N}(\|\rho\|_{H_t^{s_1} H_x^{q+s_2}}^2 + \|u\|_{H_t^{s_1} H_x^{q+s_2}}^2), \qquad (27)$$

*for $s_1, s_2 > 3/2$, where $\mathcal{M}_\psi$ is the coefficient from (14), later defined as $\mathcal{M}_\psi$ in (20).*

- *For any $\delta \in (0, 1)$ with probability $1 - \delta$ over the network parameters, the PINN training loss can be bounded as:*

$$\mathcal{J}_{PDE}(\mathbf{v}_N) \leq \frac{2(L+1)(L + \|u\|_{W_{t,x}^{1,\infty}}^2)\mathcal{M}_\psi}{N\delta}(\|\rho\|_{H_t^3 H_x^3}^2 + (\mu + 1)\|u\|_{H_t^3 H_x^4}^2), \qquad (28)$$

*if the sampled network parameters are such that $\|\rho_N\|_{W_{t,x}^{1,\infty}}^2 + \|u_N\|_{W_{t,x}^{1,\infty}}^2 \leq L < +\infty$.*

**Remark 2.** *The choice $k \geq 5$ ensures that the norms on the right-hand side of (28) are finite. With this regularity, one can also show that the boundary/initial condition residuals can be controlled similarly to $\mathcal{J}_{PDE}$, using trace inequalities.*

## 5 NUMERICAL ILLUSTRATIONS

In this section, we provide numerical experiments to validate the obtained bounds by studying the effect of network width $N$ on the error in practice. We consider the Porous Medium Equation (PME) in dimensions $d = 1, ..., 5$ and the compressible Navier-Stokes equations in $d = 1$.

### 5.1 EXPERIMENTS FOR PME

We consider the PME with $m = 2$. An exact, self-similar solution to the PME is the Barenblatt-Kompaneets-Zeldovich solution

$$u(t, x) = \frac{1}{t^\alpha}\left(b - \frac{m-1}{2m}\beta\frac{\|x\|^2}{t^{2\beta}}\right)_+^{\frac{1}{m-1}}, \qquad (29)$$

where $\|\cdot\|$ is the $\ell^2$ norm, $(\cdot)_+$ is the positive part and $\alpha = \frac{d}{d(m-1)+2}$ and $\beta = \frac{1}{d(m-1)+2}$. This solution is compactly supported but not differentiable at the edges of the support, which causes difficulty for numerical schemes. We perform two types of experiments. First, we investigate the effect of the network width $N$ on the relative error between the network and the solution $u$. This allows us to validate the convergence rate obtained in our theoretical results. Secondly, in order to assess the quality of the obtained solutions, we also compare performance of the randomised architecture against traditional PINN architectures.

### 5.1.1 EFFECT OF NETWORK WIDTH ON THE ERROR

Theorem 1 shows that for given $u : (0, T) \times \mathbb{R}^d \to \mathbb{R}$ in some Sobolev space, a RaNN $\hat{u}_N$ of width $N$ is able to achieve relative error $\mathcal{R}(u, \hat{u}) = \|\hat{u} - u\|_{L_{t,x}^2}/\|u\|_{H_t^{s_1} H_x^{s_2}}$ bounded by $C_{d,\Omega} N^{-1/2}$.

To validate this convergence rate in practice, we train a RaNN to approximate the PME solution (29) in dimensions $d = 1, ..., 5$. In each dimension, we take a set of widths $N \in \{N_1, ..., N_k\}$, train the network for each width and plot the relative error of the final network against the true solution. For each dimension $d$ and width $N$, we sample $M = 10N$ points ($M \gg N$ to ensure the problem remains well-posed) with a mixed strategy; 50% of the points are sampled uniformly on $(0, 1)^d$ and 50% are sampled uniformly on $[0.2, 0.8]^d$, which is a box focused on the initial support of the solution. Then we find weights $\mathbf{W} = \{W_i\}_{i=1}^N$ minimising the Ridge regression loss and evaluate the relative error between RaNN approximation and true solution.

The results for $d = 1, 4$ can be seen in Figure 1. The cases $d = 2, 3, 5$ are included in Figures 3, 4 in Appendix B.1.1. The key observation here is that the RaNN error points lie below or close to the $C/\sqrt{N}$ curve, which supports the upper bound of Theorem 1. Note that the sampling error becomes increasingly difficult to handle in higher dimensions, due to computational constraints.

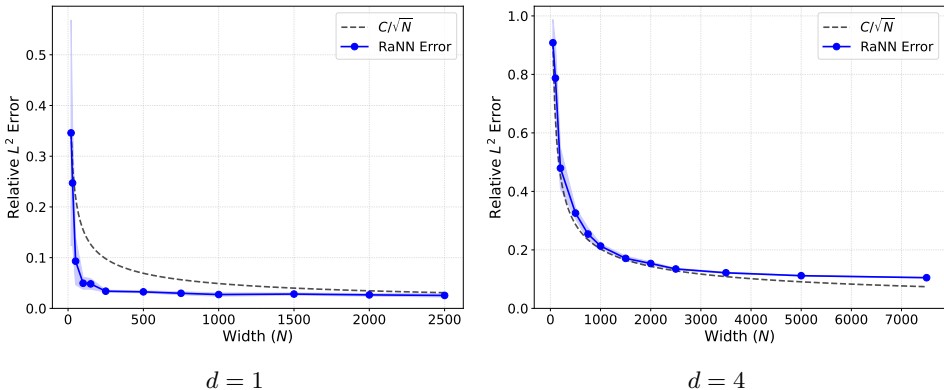

$$d = 1 \qquad\qquad\qquad d = 4$$

Figure 1: Approximation error of RaNNs of varying width for solving PMEs in dimensions $d = 1, 4$. The shaded band indicates the region within one standard deviation of the mean relative $L^2$ error.

### 5.1.2 COMPARISON BETWEEN RaNNs AND PINNs

In order to assess the quality of the obtained solutions, we also compare performance of the randomised architecture against traditional PINN architectures. We carry out simulations in dimensions $1, \ldots, 5$ using three different network architectures. The first is a RaNN with a Fourier embedding layer (see Tancik et al. (2020)). The second is a PINN which has the same architecture but where all weights/biases are trainable (denoted PINN (A)). The third is a more traditional PINN with four hidden layers and a Fourier embedding layer (denoted PINN (B)). The width of each hidden layer of PINN (B) is chosen so that the total number of parameters align with that of RaNN, which has a width of $N = 2500d$ for $d = 1, 2, 3$ and 7500 for $d = 4, 5$.

We record the relative $L^2$ error $\mathcal{X}$ of the trained solution against the true solution (29), over $(0, T) \times (0, 1)$, using $20,000$ collocation points. We also measure the relative error $\mathcal{X}_T$ between the network and true solution at the final time $T$. In each dimension the network is trained for five runs and we record the average value of $\mathcal{X}$ and $\mathcal{X}_T$ for these five runs. A compact version of the results is given in Table 1. Further experimental details and full results are given in Appendix B.1.2. Additional experiments for RaNNs without Fourier layers are provided in Appendix B.1.3.

| $d$ | Metric | PINN (A) | PINN (B) | RaNN |
|---|---|---|---|---|
| 1 | $\mathcal{X} := \|u - u_N^W\|_{L^2_{t,x}}/\|u\|_{L^2_{t,x}}$ | $7.09 \times 10^{-2}$ | $6.88 \times 10^{-2}$ | $6.41 \times 10^{-2}$ |
|  | Time (mean) | 101s | 164s | 68s |
|  | # Trainable params. | 6251 | 2665 | 2500 |
| 2 | $\mathcal{X}$ | $1.08 \times 10^{-1}$ | $1.25 \times 10^{-1}$ | $1.00 \times 10^{-1}$ |
|  | Time (mean) | 208s | 152s | 86s |
|  | # Trainable params. | 15001 | 4899 | 5000 |
| 3 | $\mathcal{X}$ | $1.24 \times 10^{-1}$ | $2.83 \times 10^{-1}$ | $1.18 \times 10^{-1}$ |
|  | Time (mean) | 579s | 187s | 107s |
|  | # Trainable params. | 26251 | 7451 | 7500 |
| 4 | $\mathcal{X}$ | $1.60 \times 10^{-1}$ | $3.48 \times 10^{-1}$ | $1.68 \times 10^{-1}$ |
|  | Time (mean) | 1034s | 225s | 138s |
|  | # Trainable params. | 30001 | 8189 | 7500 |
| 5 | $\mathcal{X}$ | $5.07 \times 10^{-1}$ | $5.33 \times 10^{-1}$ | $3.78 \times 10^{-1}$ |
|  | Time (mean) | 1395s | 326s | 168s |
|  | # Trainable params. | 33751 | 10024 | 7500 |

Table 1: Error norms and computational time for varying $d$ and $N$ values.

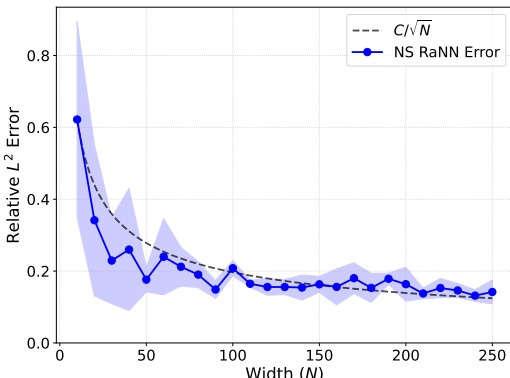

Figure 2: Approximation error of RaNNs of varying width $N$ for solving the compressible Navier-Stokes system on $(0, 1) \times [-5, 5]$. The shaded band indicates the region within one standard deviation of the mean relative $L^2$ error.

## 5.2 EXPERIMENTS FOR COMPRESSIBLE NAVIER-STOKES

We now turn to the one-dimensional compressible Navier-Stokes system, given by (22). As baseline, we consider the travelling shock-wave solutions considered by Dalibard & Perrin (2020), where the pressure $p_\epsilon(v) = \epsilon/(v-1)^\gamma$ for $\gamma > 0$ is taken, where $v = 1/\rho$. The travelling wave solutions can be obtained by taking the ansatz $(v, u)(t, x) = (\mathfrak{v}, \mathfrak{u})(x - st)$, where $s$ is the shock speed. This reduces the PDE to an ODE for $\mathfrak{v}$. The velocity $\mathfrak{u}$ can then be obtained from the relationship $\mathfrak{v} = -s\mathfrak{u}$ which follows from the conservation of mass.

For our experiment, we consider the domain $(0, T) \times (-5, 5)$ with $T = 1.0, \mu = 1, \epsilon = 10^{-3}, \gamma = 2$. We then compute RaNN approximations $(\mathfrak{v}_N, \mathfrak{u}_N)$ for different widths $N$ and measure the relative error to the baseline solution $(\mathfrak{v}, \mathfrak{u})$. The results can be seen in Figure 2, which shows that the errors are close to the $C/\sqrt{N}$ curve, in support of the upper bound of Theorem 1. Further experimental details and a visual depiction of the travelling-wave solution can be found in Appendix B.2.

## 6 CONCLUSION

In this work, we have shown that neural networks with randomly generated hidden weights (RaNNs) are capable of efficiently approximating functions that belong to time-dependent Sobolev spaces. Theorem 1 in particular demonstrates that the rate of convergence is independent of dimension $d$, which has important consequences for non-linear PDEs. To obtain this result, we used a ridgelet space representation of Sobolev functions and established a connection between ridgelet space and Sobolev space (Lemma 1).

We then demonstrated the utility of Theorem 1 by deriving error bounds on the residual loss and approximation error for solutions to two representative PDEs. Finally, we carried out a series of numerical experiments in Section 5 which validated the $N^{-1/2}$ decay rate asserted by Theorem 1.

Future work may investigate whether our results can be extended to solutions which are less smooth than those considered here. Moreover, our results assume a heavy-tailed weight distribution, while numerical experiments indicate that the same rate also holds for Gaussian weights. It would also be interesting to see whether the constant of proportionality (appearing in (14)) can be improved, either using the theory of ridgelet transforms or an alternative representation. It will also be important to explore whether RaNNs are prone to the same spectral bias issues that PINNs face, especially for complex PDEs such as compressible Navier-Stokes in the turbulent regime.

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

## A PROOFS

### A.1 PROOF OF PROPOSITION 1

*Proof.* Fix $m \geq 0$ and an arbitrary $\psi \in \mathcal{S}(\mathbb{R})$. Using the definition of the ridgelet transform from Sonoda & Murata (2017), we have for any $s \geq 0$,

$$\mathcal{R}_\psi u(\tau, \mathbf{a}, b) := \int_{\mathbb{R}^{d+1}} u(t, \mathbf{x}) \psi(\tau t + \mathbf{a} \cdot \mathbf{x} - b) \|(\tau, \mathbf{a})\|^s \, dt d\mathbf{x}. \tag{30}$$

Note that the dual ridgelet transform $R_\eta^\dagger T$ of $T : \mathbb{R}^{d+2} \to \mathbb{R}$ with respect to $\eta : \mathbb{R} \to \mathbb{R}$ is defined as

$$\mathcal{R}_\eta^\dagger T(t, \mathbf{x}) := \int_{\mathbb{R}^{d+2}} T(\tau, \mathbf{a}, b) \eta(\tau t + \mathbf{a} \cdot \mathbf{x} - b) \|(\tau, \mathbf{a})\|^{-s} d\tau d\mathbf{a} db \tag{31}$$

We will take $s = 0$ in the above definitions. Recall that $\sigma \in \mathcal{S}_0'(\mathbb{R})$ is a fixed activation. Theorem 5.6 of Sonoda & Murata (2017) says that if we can find a function $\psi$ so that $K_{\psi,\sigma} \in (0, \infty)$ then the reconstruction formula $u(t, \mathbf{x}) = \frac{1}{K_{\psi,\sigma}} R_\sigma^\dagger R_\psi u(t, \mathbf{x})$ holds. Therefore, to finish the proof we need to find $\psi$ so that $K_{\psi,\sigma} \in (0, +\infty)$ and $|\widehat{\psi}(\omega)| \leq C|\omega|^m$ for $|\omega| < 1$ (condition (8)). We will make use of Corollary 5.5 of Sonoda & Murata (2017), which says that if $\zeta^\beta \widehat{\sigma}(\zeta) \in C(-\delta, \delta)$ for some $\delta > 0, \beta \in \mathbb{N}$ and $\psi_0 \in \mathcal{S}(\mathbb{R})$ is such that

$$J := \int_{\mathbb{R}} \zeta^\beta \overline{\widehat{\psi_0}(\zeta)} \widehat{\sigma}(\zeta) \, d\zeta \neq 0, \tag{32}$$

then

$$\psi = \Lambda^d \psi_0^{(\beta)}$$

is admissible with $\sigma$, where $\Lambda^d$ is the backprojection filter that satisfies

$$\widehat{\Lambda^d F}(\mathbf{u}, \omega) = i^d |\omega|^d \widehat{F}(\mathbf{u}, \omega). \tag{33}$$

We let $G(z) := \exp(-z^2/2)$ be the standardised gaussian and take

$$\psi_0(z) := \frac{d^{2n}}{dz^{2n}} G(z), \tag{34}$$

where $n$ is an arbitrary positive integer. Then since we assume $\sigma \in \mathcal{T}_k$ (see Definition 2), we have $\zeta^\beta \widehat{\sigma}(\zeta) \in C(-\delta, \delta)$ for some $\delta > 0$ and

$$J = \int_{\mathbb{R}} \zeta^\beta \widehat{G^{(2n)}}(\zeta) \, \widehat{\sigma}(\zeta) \, d\zeta = \sqrt{2\pi}(-1)^n \int_{\mathbb{R}} \zeta^{2n+\beta} G(\zeta) \, \widehat{\sigma}(\zeta) \, d\zeta \neq 0, \tag{35}$$

where we have used $\widehat{G}(\zeta) = \sqrt{2\pi} G(\zeta)$. It is important to observe that we can take $n \in \mathbb{Z}^+$ to be as large as we like in this construction, since this is required by the second point in Definition 2 (of $\mathcal{T}_k$).

Thus, using Corollary 5.5 of Sonoda & Murata (2017) we can say that $\psi = \Lambda^d \psi_0^{(\beta)}$ and $\sigma$ are jointly admissible. We will normalise $\psi$ and therefore we can assume $K_{\psi,\sigma} = 1$. Applying the reconstruction formula (Theorem 5.6 of Sonoda & Murata (2017)) gives

$$u(t, \mathbf{x}) = R_\sigma^\dagger R_\psi u(t, \mathbf{x})$$
$$= \int_{\mathbb{R}} \int_{\mathbb{R}^d} \int_{\mathbb{R}} R_\psi u(\tau, \mathbf{a}, b) \sigma(\tau t + \mathbf{a} \cdot \mathbf{x} - b) db d\mathbf{a} d\tau. \tag{36}$$

Lastly, we need to verify that $A_{\psi,m} < +\infty$. Using the property of the backfilter from (33), we have

$$\widehat{\psi}(\omega) = i^d |\omega|^d \widehat{\psi_0'} = -i^{d+1} \omega |\omega|^{d+1} \widehat{\psi_0}$$
$$= \sqrt{2\pi} \, i^{d+2n+1} \omega |\omega|^{d+2n} G(\omega). \tag{37}$$

Taking the absolute value gives $|\widehat{\psi}(\omega)| = \sqrt{2\pi} |\omega|^{d+2n+1} G(\omega)$. Then choosing $n$ so that $d+2n+1 > m$ (e.g. $n > m - d - 1$), we have $|\widehat{\psi}(\omega)| \leq \sqrt{2\pi} |\omega|^m$ for $|\omega| < 1$. $\qquad \square$

**Remark 3** (Proving admissibility of $\tanh$, $\cos$ and sigmoid)**.** *We show here that each of these activations belong to $\mathcal{T}_k$ for $k \geq 0$. Firstly, they each belong to $\mathcal{S}_0'(\mathbb{R})$; this is explicitly mentioned in Section 6.1 of Sonoda & Murata (2017). In the case of $\sigma = \tanh$, we have $\widehat{\tanh}(\zeta) = -i\pi / \sinh(\pi\zeta/2)$ so $\zeta\widehat{\sigma}(\zeta)$ is continuous around 0, and*

$$J_\sigma = \sqrt{2\pi}(-1)^{n+1} i\pi \int_{\mathbb{R}} \frac{\zeta^{2n+1} G(\zeta)}{\sinh(\frac{\pi\zeta}{2})} \, d\zeta \neq 0. \tag{38}$$

*Furthermore, $\tanh$ and each of its derivatives are bounded uniformly. Therefore $\tanh \in \mathcal{T}_k$ for any $k \geq 0$. For $\sigma = \cos$, $\zeta\widehat{\cos}(\zeta) = \delta(\zeta + 1) - \delta(\zeta - 1)$ which is continuous (in fact, zero) in a neighbourhood of 0, and satisfies*

$$J_\sigma = \sqrt{2\pi}(-1)^n \zeta^{2n+1} \widehat{G}(\zeta)|_{-1}^1 = \sqrt{2\pi}(-1)^n \left[ G(1) + G(-1) \right] \neq 0. \tag{39}$$

*Therefore it also belongs to $\mathcal{T}_k$ for all $k \geq 0$. For $\sigma = $ sigmoid, the argument is similar; $\widehat{\sigma} = -i\pi \operatorname{csch}(\pi\zeta) + i\pi\delta(\zeta)$, so $\zeta\widehat{\sigma}(\zeta)$ is continuous around the origin and $J_\sigma \neq 0$. Each of its derivatives are also uniformly bounded.*

## A.2 SKETCHING THE PROOF OF LEMMA 1

*Proof (Outline).* **Step 1: Plancherel in** $b$. We apply the Plancherel formula in $b$ to get

$$I = \int_{\mathbb{R}^{d+2}} |\widehat{\mathcal{R}_\psi u}(\tau, \mathbf{a}, \omega)|^2 (1 + |\tau|^2)^p (1 + \|\mathbf{a}\|^2)^q \, d\tau d\mathbf{a} d\omega$$

$$+ \int_{\mathbb{R}^{d+2}} |\partial_\omega \widehat{\mathcal{R}_\psi u}(\tau, \mathbf{a}, \omega)|^2 (1 + |\tau|^2)^p (1 + \|\mathbf{a}\|^2)^q \, d\tau d\mathbf{a} d\omega \tag{40}$$

Then we show using the definition of the Fourier Transform that $\widehat{R_\psi u}(\omega) = \widehat{u}(\tau\omega, \mathbf{a}\omega)\widehat{\psi}(-\omega)$, so $I$ can be split up as

$$I \le \int_{\mathbb{R}} \int_{\mathbb{R}^d} \int_{\mathbb{R}} |\widehat{u}(\tau\omega, \mathbf{a}\omega)\widehat{\psi}(-\omega)|^2 (1 + |\tau|^2)^p (1 + \|\mathbf{a}\|^2)^q \, d\omega d\tau d\mathbf{a}$$

$$+ \int_{\mathbb{R}} \int_{\mathbb{R}^d} \int_{\mathbb{R}} |\partial_\omega \left( \widehat{u}(\tau\omega, \mathbf{a}\omega)\widehat{\psi}(-\omega) \right)|^2 (1 + |\tau|^2)^p (1 + \|\mathbf{a}\|^2)^q \, d\omega d\tau d\mathbf{a} =: I_1 + I_2 \tag{41}$$

The second term $I_2$ is the main obstacle in the proof.

**Step 2: Change of variables.** Performing a change of variables $s = \tau\omega, \xi = \mathbf{a}\omega$ allows us to estimate $I_2$ as

$$I_2 \le \int_{\mathbb{R}} \int_{\mathbb{R}^d} \int_{\mathbb{R}} |\widehat{u}(s, \xi)\partial_\omega \widehat{\psi}(-\omega)|^2 (1 + (\frac{s}{\omega})^2)^p (1 + (\frac{\|\xi\|}{\omega})^2)^q |\omega|^{-(d+1)} \, d\omega ds d\xi$$

$$+ \int_{\mathbb{R}} \int_{\mathbb{R}^d} \int_{\mathbb{R}} |\frac{1}{\omega}(s\partial_s + \xi\nabla_\xi)\widehat{u}(s, \xi)\widehat{\psi}(-\omega)|^2 (1 + (\frac{s}{\omega})^2)^p (1 + (\frac{\|\xi\|}{\omega})^2)^q |\omega|^{-(d+1)} \, d\omega ds d\xi$$

$$=: I_{2A} + I_{2B} \tag{42}$$

**Step 3: Estimating $I_{2A}$ and $I_{2B}$.** Here we show that one can find $M$ large enough (but still finite) so that with the corresponding $\psi$ generated from Proposition 1, the integrals $I_{2A}, I_{2B}$ are both bounded by constants depending on the $L^2$ norms of $\psi$. Most of the difficulty lies in $I_{2B}$ due to the derivatives that appear. Nonetheless, using properties of the Fourier transform and Fubini to exchange the order of integration, $I_2$ can be estimated as

$$I_{2B} \le C_{\psi_2} \int_{\mathbb{R}^d} |\widehat{(-it)u}|^2 (s, \xi)(1 + |s|^2)^{p+r}(1 + \|\xi\|^2)^q \, d\xi ds$$

$$+ C_{\psi_3} \int_{\mathbb{R}^d} |\widehat{(-i\mathbf{x})u}|^2 (s, \xi)(1 + |s|^2)^p (1 + \|\xi\|^2)^{q+r} \, d\xi ds, \tag{43}$$

where

$$C_{\psi_2} = \int_{\mathbb{R}} |\partial_\omega \widehat{\psi}(-\omega)|^2 \max(1, |\omega|^{-(2p+2q)})|\omega|^{-(d+1)} \, d\omega.,$$

$$C_{\psi_3} = \int_{\mathbb{R}} |\widehat{\psi}(-\omega)|^2 \max(1, |\omega|^{-(2p+2q)})|\omega|^{-(d+3)} \, d\omega. \tag{44}$$

Then by definition of the space $H^p(\mathbb{R}; H^q(\mathbb{R}^d))$ and the compact support of $u$, we get

$$I_{2B} \le C_{\psi_3}(2T + 2R)(\|u\|^2_{H_t^{p+1}H_x^q} + \|u\|^2_{H_t^p H_x^{q+1}}), \tag{45}$$

where $C_{\psi_1}$ is a constant similar in form to $C_{\psi_3}$, and appears due to $I_1$ (which we did not look at in this outline). Our final job is to show that we can construct $\psi$ so that each of $C_{\psi_1}, C_{\psi_2}, C_{\psi_3}$ can be suitably bounded. This is done for each constant by separating $|\omega| < 1$ and $|\omega| \ge 1$. On $|\omega| < 1$ we take advantage of Proposition 1 which allows us to choose $\psi$ with enough vanishing moments at $0$ to ensure finiteness. On $|\omega| \ge 1$ we use $\psi \in \mathcal{S}(\mathbb{R})$. This is the main idea. The full details of the proof are deferred to the appendix. $\square$

### A.3 PROOF OF LEMMA 1

*Proof.* **Step 1: Plancherel in** $b$**.** Firstly, let's note that using the Plancherel formula in $b$ we have

$$\int_{\mathbb{R}} |\mathcal{R}_\psi u(\tau, \mathbf{a}, b)|^2 (1 + b^2) \, db = \int_{\mathbb{R}} |\mathcal{R}_\psi u(\tau, \mathbf{a}, b)|^2 \, db + \int_{\mathbb{R}} |b \mathcal{R}_\psi u(\tau, \mathbf{a}, b)|^2 \, db$$

$$= \int_{\mathbb{R}} |\widehat{\mathcal{R}_\psi u}(\tau, \mathbf{a}, \omega)|^2 \, d\omega + \int_{\mathbb{R}} |\partial_\omega \widehat{\mathcal{R}_\psi u}(\tau, \mathbf{a}, \omega)|^2 \, d\omega. \tag{46}$$

Then

$$I = \int_{\mathbb{R}^{d+2}} |\widehat{\mathcal{R}_\psi u}(\tau, \mathbf{a}, \omega)|^2 (1 + |\tau|^2)^p (1 + \|\mathbf{a}\|^2)^q \, d\tau d\mathbf{a} d\omega$$

$$+ \int_{\mathbb{R}^{d+2}} |\partial_\omega \widehat{\mathcal{R}_\psi u}(\tau, \mathbf{a}, \omega)|^2 (1 + |\tau|^2)^p (1 + \|\mathbf{a}\|^2)^q \, d\tau d\mathbf{a} d\omega \tag{47}$$

We now find an expression for $\widehat{\mathcal{R}_\psi u}$. Using the definition of the Fourier transform and the ridgelet transform,

$$\widehat{R_\psi u}(\omega) = \int_{\mathbb{R}} e^{-i\omega b} \, \mathcal{R}_\psi u \, db$$

$$= \int_{\mathbb{R}} e^{-i\omega b} \left( \int_{\mathbb{R}^d} \int_{\mathbb{R}} u(t, \mathbf{x}) \psi(\tau t + \mathbf{a} \cdot \mathbf{x} - b) \, dt dx \right) db \tag{48}$$

$$= \int_{\mathbb{R}^d} \int_{\mathbb{R}} u(t, \mathbf{x}) \left( \int_{\mathbb{R}} e^{-i\omega b} \psi(\tau t + \mathbf{a} \cdot \mathbf{x} - b) \, db \right) dt dx,$$

where we have also used Fubini to exchange the order of the integrals. This is valid since we assume $u$ is compactly supported, and therefore $u\psi$ is integrable on $\mathbb{R}^{d+1}$. By a change of variables $p = \tau t + \mathbf{a} \cdot \mathbf{x} - b$ the inner integral is equivalent to $e^{-i\omega(\tau t + \mathbf{a} \cdot \mathbf{x})} \int_{\mathbb{R}} e^{i\omega p} \psi(p) \, dp = e^{-i\omega(\tau t + \mathbf{a} \cdot \mathbf{x})} \widehat{\psi}(-\omega)$. Therefore

$$\widehat{R_\psi u}(\omega) = \widehat{\psi}(-\omega) \int_{\mathbb{R}^d} u(t, \mathbf{x}) e^{-i\omega(\tau t + \mathbf{a} \cdot \mathbf{x})} \, dt d\mathbf{x}$$

$$= \widehat{u}(\tau\omega, \mathbf{a}\omega) \widehat{\psi}(-\omega). \tag{49}$$

Inserting this into (47), we have

$$I = \int_{\mathbb{R}} \int_{\mathbb{R}^d} \int_{\mathbb{R}} |\widehat{u}(\tau\omega, \mathbf{a}\omega) \widehat{\psi}(-\omega)|^2 (1 + |\tau|^2)^p (1 + \|\mathbf{a}\|^2)^q \, d\omega d\tau d\mathbf{a}$$

$$+ \int_{\mathbb{R}} \int_{\mathbb{R}^d} \int_{\mathbb{R}} |\partial_\omega \left( \widehat{u}(\tau\omega, \mathbf{a}\omega) \widehat{\psi}(-\omega) \right)|^2 (1 + |\tau|^2)^p (1 + \|\mathbf{a}\|^2)^q \, d\omega d\tau d\mathbf{a} \tag{50}$$

$$=: I_1 + I_2.$$

**Step 2: Change of variables.** We now introduce the change of variables $s = \tau\omega, \xi = \mathbf{a}\omega$. The corresponding jacobian is $d\mathbf{a} d\tau = |\omega|^{-(d+1)} d\xi ds$ and the operator changes to

$$\partial_\omega|_{(\tau, \mathbf{a})} = \partial_\omega|_{(s, \xi)} + \frac{1}{\omega} (s\partial_s + \xi \nabla_\xi)|_{(s, \xi)} =: \partial_\omega|_{(s, \xi)} + \mathcal{D}. \tag{51}$$

Therefore, separating the $\omega$ terms using Fubini,

$$I_1 = \int_{\mathbb{R}^{d+2}} |\widehat{u}(s, \xi)|^2 |\widehat{\psi}(-\omega)|^2 (1 + |s/\omega|^2)^p (1 + (\|\xi\|/\omega)^2)^q |\omega|^{-(d+1)} \, d\omega ds d\xi$$

$$\leq C_{\psi_1} \int_{\mathbb{R}^{d+1}} |\widehat{u}(s, \xi)|^2 (1 + |s|^2)^p (1 + \|\xi\|^2)^q \, ds d\xi, \tag{52}$$

where

$$C_{\psi_1} = \int_{\mathbb{R}} |\widehat{\psi}(-\omega)|^2 \max(1, |\omega|^{-2p}) \max(1, |\omega|^{-2q}) |\omega|^{-(d+1)} \, d\omega. \tag{53}$$

We will show the boundedness of this constant at the end of the proof. For $I_2$,

$$I_2 \leq \int_{\mathbb{R}} \int_{\mathbb{R}^d} \int_{\mathbb{R}} |\widehat{u}(s,\xi) \partial_\omega \widehat{\psi}(-\omega)|^2 (1 + (\frac{s}{\omega})^2)^p (1 + (\frac{\|\xi\|}{\omega})^2)^q |\omega|^{-(d+1)} \, d\omega ds d\xi$$

$$+ \int_{\mathbb{R}} \int_{\mathbb{R}^d} \int_{\mathbb{R}} |\frac{1}{\omega}(s\partial_s + \xi\nabla_\xi)\widehat{u}(s,\xi)\widehat{\psi}(-\omega)|^2 (1 + (\frac{s}{\omega})^2)^p (1 + (\frac{\|\xi\|}{\omega})^2)^q |\omega|^{-(d+1)} \, d\omega ds d\xi$$

$$=: I_{2A} + I_{2B} \tag{54}$$

**Step 3: Estimating $I_{2A}$ and $I_{2B}$.** First note that for any $\mathbf{x} \in \mathbb{R}^d$ and $k > 0$,

$$(1 + \frac{\|\mathbf{x}\|^2}{|\omega|^2})^k \leq \max(1, |\omega|^{-2k})(1 + \|\mathbf{x}\|^2)^k. \tag{55}$$

Therefore, for $I_{2A}$ we can separate the $\omega$ dependence and write

$$I_{2A} \leq C_{\psi_2} \int_{\mathbb{R}^{d+1}} |\widehat{u}(s,\xi)|^2 (1 + s^2)^p (1 + \|\xi\|^2)^q ds d\xi, \tag{56}$$

where

$$C_{\psi_2} = \int_{\mathbb{R}} |\partial_\omega \widehat{\psi}(-\omega)|^2 \max(1, |\omega|^{-(2p+2q)})|\omega|^{-(d+1)} \, d\omega. \tag{57}$$

We will consider the boundedness of this constant at the end. For $I_{2B}$, we can estimate the $(s,\xi)$ part as

$$\int_{\mathbb{R}^{d+1}} |\frac{1}{\omega}(s\partial_s + \xi\nabla_\xi)\widehat{u}(s,\xi)|^2 \, ds d\xi \leq 2|\omega|^{-2} \left( \int_{\mathbb{R}^{d+1}} |s\partial_s \widehat{u}(s,\xi)|^2 + \int_{\mathbb{R}^{d+1}} |\xi\nabla_\xi \widehat{u}(s,\xi)|^2 \right)$$

$$= 2|\omega|^{-2} \left( \int_{\mathbb{R}^{d+1}} |s\widehat{(-it)u}(s,\xi)|^2 + \int_{\mathbb{R}^{d+1}} |\xi\widehat{(-i\mathbf{x})u}(s,\xi)|^2 \right)$$

$$\leq 2|\omega|^{-2} \left( \int_{\mathbb{R}^{d+1}} |\widehat{(-it)u}(s,\xi)|^2(1+s^2) + \int_{\mathbb{R}^{d+1}} |\widehat{(-i\mathbf{x})u}(s,\xi)|^2(1+\|\xi\|^2) \right), \tag{58}$$

where we used $s^2 \leq 1 + s^2$ to obtain the final line. Thus we have

$$I_{2B} \leq C_{\psi_3} \int_{\mathbb{R}^{d+1}} |\widehat{(-it)u}|^2(s,\xi)(1 + |s|^2)^{p+1}(1 + \|\xi\|^2)^q \, d\xi ds$$

$$+ C_{\psi_3} \int_{\mathbb{R}^{d+1}} |\widehat{(-i\mathbf{x})u}|^2(s,\xi)(1 + |s|^2)^p(1 + \|\xi\|^2)^{q+1} \, d\xi ds, \tag{59}$$

where

$$C_{\psi_3} = 2 \int_{\mathbb{R}} |\widehat{\psi}(-\omega)|^2 \max(1, |\omega|^{-(2p+2q)})|\omega|^{-(d+3)} \, d\omega. \tag{60}$$

We will consider this constant at the end of the proof. Next, recall the following equivalence

$$\int_{\mathbb{R}^{d+1}} |\widehat{u}(s,\xi)|^2 (1 + |s|^2)^p (1 + \|\xi\|^2)^q \, d\xi ds = \|u\|^2_{H^p(\mathbb{R}; H^q(\mathbb{R}^d))}. \tag{61}$$

This gives us (using the compact support of $u$)

$$I \leq (C_{\psi_1} + C_{\psi_2})\|u\|_{H_t^p H_x^q} + C_{\psi_3}(\||t|u\|^2_{H_t^{p+1}H_x^q} + \||\mathbf{x}|u\|^2_{H_t^p H_x^{q+1}})$$

$$\leq (C_{\psi_1} + C_{\psi_2})\|u\|_{H_t^p H_x^q} + C_{\psi_3}(2T + 2R)\|u\|^2_{H_t^{p+1}H_x^{q+1}} \tag{62}$$

$$\leq (C_{\psi_1} + C_{\psi_2} + C_{\psi_3}(2T + 2R))\|u\|^2_{H_t^{p+1}H_x^{q+1}}.$$

It remains to bound the constants $C_{\psi_1}$, $C_{\psi_2}$ and $C_{\psi_3}$. Starting with the most recent $C_{\psi_3}$, we consider $|\omega| < 1$ and $|\omega| \geq 1$ separately. On $|\omega| < 1$, we have

$$C_{\psi_3} = 2 \int_{-1}^1 |\widehat{\psi}(-\omega)|^2 \max(1, |\omega|^{-(2p+2q)})|\omega|^{-(d+3)} \, d\omega \leq 2 \int_{-1}^1 |\widehat{\psi}(-\omega)|^2 |\omega|^{-(2p+2q+d+3)} \, d\omega,$$

so letting $M = (2p + 2q + d + 3)/2$ and choosing $\psi$ as per the construction of Proposition 1 (see (37)), we have

$$|\widehat{\psi}(\omega)| \leq \sqrt{2\pi}|\omega|^M \text{ for } |\omega| < 1. \tag{63}$$

This implies that the integral is bounded by $\int_{-1}^{1} 2\pi d\omega = 4\pi$. On $|\omega| \geq 1$ we can bound it by $\|\psi\|_{L^2(\mathbb{R})}^2$ (by Plancherel), and so $C_{\psi_3} \leq 8\pi + 2\|\psi\|_{L^2(\mathbb{R})}$. We can apply the same estimate to $C_{\psi_1}$ to get $C_{\psi_1} \leq 4\pi + \|\psi\|_{L^2(\mathbb{R})}^2$. For $C_{\psi_2}$, recall from (37) that $\widehat{\psi}(\omega) = \sqrt{2\pi} \, i^m \omega |\omega|^{m-1} G(\omega)$, for $m$ which we can choose arbitrarily large. A simple computation gives us on $|\omega| < 1$ that $|\partial_\omega \widehat{\psi}| \leq \sqrt{2\pi}(m+1)\omega^{m-1}G(\omega)$. Similarly, on $|\omega| \geq 1$ we get $|\partial_\omega \widehat{\psi}| \leq \sqrt{2\pi}m\omega^{m+1}G(\omega)$. Therefore on $|\omega| < 1$, by choosing the same $m = M$,

$$\int_{-1}^{1} |\partial_\omega \widehat{\psi}(-\omega)|^2 \max(1, |\omega|^{-(2p+2q)})|\omega|^{-(d+1)} \, d\omega \leq 2\pi(M+1)^2 \int_{-1}^{1} |\omega|^2 \, d\omega \tag{64}$$

$$\leq 4\pi(M+1)^2.$$

For $|\omega| > 1$, notice that $|\partial_\omega \widehat{\psi}| \leq |\omega||\widehat{\psi}(\omega)|$ and so

$$\int_{\mathbb{R}\backslash B_1(0)} |\partial_\omega \widehat{\psi}(-\omega)|^2 \max(1, |\omega|^{-(2p+2q)})|\omega|^{-(d+1)} \, d\omega \leq \int_{\mathbb{R}\backslash B_1(0)} |\omega\widehat{\psi}(\omega)|^2 \, d\omega. \tag{65}$$

and so $C_{\psi_2} \leq 4\pi(M+1)^2 + \|\psi'\|_{L^2(\mathbb{R})}^2$. This leads to

$$I \leq \left[ (4\pi + \|\psi\|_{L^2(\mathbb{R})})(1 + 4T + 4R) + 4\pi(M+1)^2 + \|\psi'\|_{L^2(\mathbb{R})}^2 \right] \|u\|_{H_t^{p+1}H_x^{q+1}}^2, \tag{66}$$

where $M = (2p + 2q + d + 3)/2$. $\qquad\square$

## A.4 PROOF OF THEOREM 1

*Proof.* **Step 1: extension of $u$ to $\mathbb{R} \times \mathbb{R}^d$.** In order to make use of the previous results, we smoothly extend $u$ to the full space $\mathbb{R} \times \mathbb{R}^d$ using cut-off functions. More precisely, we define $\tilde{u} = \eta\chi u$, where $\chi \in C_c^\infty(\mathbb{R}^d), \eta \in C_c^\infty(\mathbb{R})$ with $\chi \equiv 1$ on $D$, $\chi = 0$ outside of $[-2R, 2R]^d$ and $\eta \equiv 1$ on $[0, T]$, $\eta = 0$ outside of $[-2T, 2T]$, so that $\|\tilde{u}\|_{H_t^p H_x^q} \leq C_\Omega \|u\|_{H_t^p H_x^q}$, where $C_\Omega$ is a constant dependent on $p, q, d$ and the domain. Such an extension is known to exist (see e.g. Chapter VI of Stein (1970)). Then from (7), $\tilde{u}$ can be represented as

$$\tilde{u}(t, x) = \int_{\mathbb{R}} \int_{\mathbb{R}^d} \int_{\mathbb{R}} (\mathcal{R}_\psi \tilde{u})(\tau, \mathbf{a}, b)\sigma(\tau t + \mathbf{a} \cdot \mathbf{x} - b) \, d\tau d\mathbf{a} db$$

$$= \int_{\mathbb{R}} \int_{\mathbb{R}^d} \int_{\mathbb{R}} \frac{(\mathcal{R}_\psi \tilde{u})(\tau, \mathbf{a}, b)}{\pi(\tau, \mathbf{a}, b)}\sigma(\tau t + \mathbf{a} \cdot \mathbf{x} - b)\pi(\tau, \mathbf{a}, b)d\tau d\mathbf{a} db, \tag{67}$$

where $\pi : \mathbb{R}^{d+2} \to \mathbb{R}_+$ is the probability density function from (12).

**Step 2: construction of the unbiased estimator $u_N$.** Our neural network approximation of $u$ will be denoted $u_N$, and we define it as an unbiased estimator of $\tilde{u}$, i.e.

$$u_N(t, \mathbf{x}) := \frac{1}{N} \sum_{i=1}^{N} \frac{R_\psi \tilde{u}(\tau_i, \mathbf{a}_i, b_i)}{\pi(\tau_i, \mathbf{a}_i, b_i)}\sigma(\tau_i t + \mathbf{a}_i \cdot \mathbf{x} - b_i) \equiv \frac{1}{N} \sum_{i=1}^{N} X_i(t, \mathbf{x}), \tag{68}$$

where $(\tau_i, \mathbf{a}_i, b_i) \sim \pi$. By construction, it is clear that $\mathbb{E}_\Theta(u_N) = \tilde{u}$. More generally, for any $0 \leq \ell \leq p$ and multi-index $\beta$ with $|\beta| \leq q$, we have

$$\mathbb{E}(\partial_t^\ell D_\mathbf{x}^\beta(u_N)) = \partial_t^\ell D_\mathbf{x}^\beta(\tilde{u}). \tag{69}$$

In order to estimate $u - u_N$, we have

$$\mathbb{E}_\Theta \left( \|\partial_t^\ell D_\mathbf{x}^\beta(\tilde{u} - u_N)\|_{L^2((0,T)\times D)}^2 \right) = \mathbb{E}_\Theta \int_{(0,T)\times D} |\partial_t^\ell D_\mathbf{x}^\beta(\tilde{u} - u_N)|^2 \, dxdt$$

$$= \int_{(0,T)\times D} \text{Var}_\Theta \left( \partial_t^\ell D_\mathbf{x}^\beta(\tilde{u} - u_N) \right) \, dxdt. \tag{70}$$

Let $Y_i(t, \mathbf{x}) := X_i(t, \mathbf{x}) - \mathbb{E}_\Theta(X_1(t, \mathbf{x}))$. Then $\mathbb{E}(Y_i) = 0$ and $\mathrm{Var}(Y_i) = \mathrm{Var}(X_i)$. Moreover, we have $u - u_N = \frac{1}{N} \sum_{i=1}^N Y_i(t, \mathbf{x})$ and therefore $\mathrm{Var}(u - u_N) = \frac{1}{N} \mathrm{Var}(Y_1(t, \mathbf{x})) = \frac{1}{N} \mathrm{Var}(X_1(t, \mathbf{x}))$. The same argument works if we replace $u - u_N$ with $\partial_t^\ell D_\mathbf{x}^\beta(u - u_N)$. As a result, we have (since $u = \tilde{u}$ on $(0, T) \times D$)

$$\mathbb{E}_\Theta \left( \|\partial_t^\ell D_\mathbf{x}^\beta(u - u_N)\|_{L^2((0,T)\times D)}^2 \right) = \mathbb{E}_\Theta \left( \|\partial_t^\ell D_\mathbf{x}^\beta(\tilde{u} - u_N)\|_{L^2((0,T)\times D)}^2 \right)$$
$$= \frac{1}{N} \int_{(0,T)\times D} \mathbb{E}_\Theta |\partial_t^\ell D_\mathbf{x}^\beta X_i|^2 \, dxdt. \tag{71}$$

**Step 3: bounding $\mathbb{E}_\Theta(|\partial_t^\ell D_\mathbf{x}^\alpha X_i|^2)$.** A direct computation gives

$$\mathbb{E}_\Theta(|X_i(t, \mathbf{x})|^2)$$
$$= C_\pi \int_{\mathbb{R}^{d+2}} |R_\psi \tilde{u}(\tau, \mathbf{a}, b)|^2 |\sigma(\tau t + \mathbf{a}\mathbf{x} - b)|^2 (1 + \tau^2)^{\lambda_\tau} (1 + \|\mathbf{a}\|^2)^{\lambda_a} (1 + b^2) \, d\tau d\mathbf{a} db \tag{72}$$
$$\leq C_\pi \|\sigma\|_\infty^2 \mathcal{L}_\psi \|\tilde{u}\|_{H^{s_1}(\mathbb{R}; H^{s_2}(\mathbb{R}^d))}^2,$$

where, since $\lambda_\tau > 1/2$ and $\lambda_a > d/2$, we have $s_1 > 3/2, s_2 > (d+2)/2$ using Lemma 1. Therefore, we have

$$\mathbb{E}_\Theta \left( \|u - u_N\|_{L^2((0,T)\times D)}^2 \right) \leq \frac{T|D|\|\sigma\|_\infty^2 \mathcal{L}_\psi}{N} \|\tilde{u}\|_{H^{s_1}(\mathbb{R}; H^{s_2}(\mathbb{R}^d))}^2. \tag{73}$$

The same argument works if we let $0 \leq \ell \leq p$ and use a non-zero multi-index $\beta = (\beta_1, ..., \beta_d)$ with $|\beta| = q$. In this case we have

$$\mathbb{E}_\Theta(|\partial_t^\ell D_\mathbf{x}^\alpha X_i(t, \mathbf{x})|^2)$$
$$\leq C_\pi \int_{\mathbb{R}^{d+2}} |R_\psi \tilde{u}(\tau, \mathbf{a}, b)|^2 |\sigma^{(p+q)}|^2 |\tau|^{2p} \|\mathbf{a}\|^{2q} (1 + \tau^2)^{\lambda_\tau} (1 + \|\mathbf{a}\|^2)^{\lambda_a} (1 + b^2) \, d\tau d\mathbf{a} db$$
$$\leq C_\pi \int_{\mathbb{R}^{d+2}} |R_\psi \tilde{u}(\tau, \mathbf{a}, b)|^2 |\sigma^{(p+q)}|^2 (1 + \tau^2)^{p+\lambda_\tau} (1 + \|\mathbf{a}\|^2)^{q+\lambda_a} (1 + b^2) \, d\tau d\mathbf{a} db. \tag{74}$$

Then using (10), we get

$$\mathbb{E}_\Theta(|\partial_t^\ell D_\mathbf{x}^\alpha X_i(t, \mathbf{x})|^2) \leq C_\pi \|\sigma^{(p+q)}\|_\infty^2 \mathcal{L}_\psi \|\tilde{u}\|_{H^{p+s_1}(\mathbb{R}; H^{q+s_2}(\mathbb{R}^d))}^2, \tag{75}$$

By (71) this implies (using $\|\tilde{u}\|_{H_t^p H_x^q} \leq C_\Omega \|u\|_{H_t^p H_x^q}$)

$$\mathbb{E}_\Theta \left( \|\partial_t^\ell D_\mathbf{x}^\beta(u - u_N)\|_{L^2((0,T)\times D)}^2 \right) \leq \frac{C_\Omega T|D|C_\pi \|\sigma^{(p+q)}\|_\infty^2 \mathcal{L}_\psi}{N} \|u\|_{H^{p+s_1}(\mathbb{R}; H^{q+s_2}(\mathbb{R}^d))}^2, \tag{76}$$

where again $s_1 > 3/2$ and $s_2 > (d+2)/2$. Summing up this estimate for each of the derivatives up to order $(p, q)$ leads to the claimed result. $\qquad \square$

## A.5 PROOF OF COROLLARY 1

*Proof.* The bound of (20) follows from an application of Theorem 1, so we now focus on bounding the PDE residual. We take $m = 2$ for simplicity, but the computation is easily generalised to any $m \geq 1$. Using $\partial_t u - \Delta(u^2) = 0$, we have

$$\mathcal{J}_{\mathrm{PDE}}(u_N) = \int_{(0,T)\times D} (\partial_t u_N - \Delta(u_N^2))^2 \, dtdx$$
$$= \int_{(0,T)\times D} (\partial_t(u_N - u) - \Delta(u_N^2 - u^2))^2 \, dtdx \tag{77}$$
$$\leq 2\|u_N - u\|_{H_t^1 L_x^2}^2 + 2 \int_{(0,T)\times D} |\Delta(u_N^2 - u^2)|^2 \, dtdx.$$

Now, Markov's inequality states that $\mathbb{P}(X > \eta) \leq \mathbb{E}[X]\frac{1}{\eta}$ for $X$ non-negative, $\eta > 0$. In particular, for any $\delta \in (0,1)$, from the bound in Theorem 1 we obtain with the choices $X = \|u - u_N\|^2_{H^p(0,T;H^q(D))}$ and $\eta = \frac{1}{\delta} \frac{C_\Omega C_\pi \|\sigma^{(p+q)}\|^2_\infty T|D|(p+q)\mathcal{L}_\psi}{N} \cdot \|u\|^2_{H^{p+s_1}(\mathbb{R};H^{q+s_2}(\mathbb{R}^d))}$ that

$$\mathbb{P}(X > \eta) \leq \mathbb{E}_\Theta \left( \|u - u_N\|^2_{H^p(0,T;H^q(D))} \right) \frac{1}{\eta} \leq \delta. \tag{78}$$

This implies that $\mathbb{P}(X \leq \eta) \geq 1 - \delta$. That is, with probability at least $1 - \delta$ it holds that $X \leq \eta = \frac{1}{\delta} \frac{C_\Omega C_\pi \|\sigma^{(p+q)}\|^2_\infty T|D|(p+q)\mathcal{L}_\psi}{N} \cdot \|u\|^2_{H^{p+s_1}(\mathbb{R};H^{q+s_2}(\mathbb{R}^d))}$. Then we have with probability $1 - \delta$ that

$$\|u_N - u\|^2_{H^1_t L^2_x} \leq \frac{\mathcal{M}_\psi}{N\delta} \|u\|^2_{H^{1+s_1}_t H^{s_2}_x}, \tag{79}$$

where $s_1 > 3/2, s_2 > (d+2)/2$. Now we note that $\Delta(u_N^2 - u^2) = (u_N + u)\Delta e_N + 2\nabla(u_N + u)\nabla e_N + e_N\Delta(u_N + u)$, where $e_N := u_N - u$. Combining this with $(a+b)^2 \leq 2(a^2 + b^2)$ and Holder's inequality,

$$\int_{(0,T)\times D} |\Delta(u_N^2 - u^2)|^2 \, dtdx \leq 2\|u_N + u\|^2_{L^\infty_{t,x}} \|\Delta(u_N - u)\|^2_{L^2_{t,x}}$$
$$+ 16\|\nabla(u_N + u)\|^2_{L^\infty_{t,x}} \|\nabla(u_N - u)\|^2_{L^2_{t,x}} \tag{80}$$
$$+ 4\|\Delta(u_N + u)\|^2_{L^\infty_{t,x}} \|u_N - u\|^2_{L^2_{t,x}}.$$

We also have

$$\|u\|_{L^\infty_{t,x}} + \|\nabla u\|_{L^\infty_{t,x}} + \|\Delta u\|_{L^\infty_{t,x}} \leq C_{emb}\|u\|_{L^\infty_t H^{2+k}_x}, \tag{81}$$

for $k > d/2$, where $C_{emb}$ is the constant arising from the Sobolev embedding $H^k_x \hookrightarrow L^\infty_x$. Then with the above Markov argument, assuming $L$ is such that the sampling of the weights satisfies $\|u_N\|_{L^\infty_{t,x}} + \|\nabla u_N\|_{L^\infty_{t,x}} + \|\Delta u_N\|_{L^\infty_{t,x}} \leq L$, we have with probability $1 - \delta$ that

$$\int_{(0,T)\times D} |\Delta(u_N^m - u^m)|^2 \, dtdx \leq 22(L + C_{emb}\|u\|^2_{L^\infty_t H^{2+k}_x})\|u_N - u\|^2_{L^2_t H^2_x}$$
$$\leq \frac{22(L + C_{emb}\|u\|^2_{L^\infty_t H^{2+k}_x})\mathcal{M}_\psi}{N\delta} \|u\|^2_{H^{s_1}_t H^{2+s_2}_x}. \tag{82}$$

All in all, we get

$$\mathcal{J}_{\text{PDE}}(u_N) \leq \frac{(22(L + C_{emb}\|u\|^2_{L^\infty_t H^{2+k}_x}) + 1)\mathcal{M}_\psi}{N\delta} \|u\|^2_{H^{s_1}_t H^{2+s_2}_x}, \tag{83}$$

with probability $1 - \delta$. Using $s_1 > 3/2$ and $s_2 > d/2 + 2$ we can estimate the leading norm by $\|u\|^2_{H^2_t H^{3+k}_x}$. For the general case $m \geq 2$, the argument can be adapted using a mean-value formula to re-write $u_N^m - u^m$ in terms of $e_N$. In this case, we get an extra constant in the coefficient of (83), $C_m$, which will be polynomial in $m$. This leads to the claimed result (21). In summary, taking initial data $u_0 \in H^1(\mathbb{R}^d)$, we can find a RaNN which approximates the corresponding solution to the PDE such that the expected loss of the PDE residual is inversely proportional to the number of random features, and the rate is independent of dimension. □

**Remark 4** (Growth factor for the leading coefficient). *We can give a more precise description of the constant in the right-hand side of* (14) *in the case where $\sigma \in \mathcal{T}_k$ (e.g.* tanh*; see Definition 2) and either $u \in H^p(\mathbb{R}; H^q(\mathbb{R}^d))$ (i.e. $u$ is already defined on the full space) or $p = q = 0$. The condition $p = q = 0$ ensures that $C_\Omega = 1$ since a zero extension suffices. In such a case, we can show that if we let $\lambda_a = \frac{d}{2} + \alpha(d))$ for a suitable choice of $\alpha$, then the coefficient in the right-hand side of* (14) *will grow at most polynomially in $d$. We will denote by $C$ a positive constant independent of dimension and by $\Gamma(\cdot)$ the Gamma function. Then note that*

$$C_\pi \leq C \int_{\mathbb{R}^d} (1 + \|\mathbf{a}\|^2)^{-\lambda_a} \, d\mathbf{a}. \tag{84}$$

*Going to spherical coordinates and letting $\lambda_a = \frac{d}{2} + \alpha$, we find*

$$C_\pi = \frac{2\pi^{d/2}}{\Gamma(d/2)} \int_0^\infty r^{d-1}(1+r^2)^{-\lambda_a} \, dr = \pi^{d/2} \frac{\Gamma(\alpha)}{\Gamma(d/2+\alpha)} \leq C\pi^{d/2} \frac{\alpha^{\alpha-1/2}}{e^\alpha}. \quad (85)$$

*Now, to bound $L_\psi$, we need an estimate for $\|\widehat{\psi}\|^2_{L^2(\mathbb{R})}$ and $\|(\widehat{\psi})'\|_{L^2(\mathbb{R})}$. We will use the construction of $\psi$ from the proof of Proposition 1. There, we construct $\psi$ with $|\widehat{\psi}(\omega)| \leq \sqrt{2\pi}|\omega|$ for $|\omega| \leq 1$ and $|\widehat{\psi}(\omega)| \leq \sqrt{2\pi}|\omega|^{d+3}G(\omega)$ for $|\omega| > 1$ (we set $n = 1$ for this proof but it can be extended to any fixed positive integer $n$). Then we have $\|\widehat{\psi}\|^2_{L^2(-1,1)} \leq 4\pi$ and so*

$$\|\widehat{\psi}\|^2_{L^2(\mathbb{R})} \leq 4\pi + \int_{|\omega|>1} \omega^{2(d+2)}e^{-\omega^2} \, d\omega$$

$$= 4\pi + \int_1^\infty u^{d+3/2}e^{-u} \, du \qquad (86)$$

$$\leq 4\pi + \Gamma(d+5/2),$$

*where we have used the substitution $u = \omega^2$. For $(\widehat{\psi})'$, we can repeat the same process to find*

$$\|(\widehat{\psi})'\|^2_{L^2(\mathbb{R})} \leq 4\pi + \Gamma(d+7/2). \quad (87)$$

*Now we consider the product $C_\pi L_\psi$. Notice that the leading term is given by $C_\pi \Gamma(d + 7/2)$, or more precisely by*

$$F(d) := \pi^{d/2} \frac{\Gamma(\alpha)}{\Gamma(d/2+\alpha)} \sqrt{\Gamma(d+7/2)}. \quad (88)$$

*We now prove that $F(d)$ grows at most polynomially in $d$. We will take $\alpha = \lambda d$ for some real number $\lambda > 0$ to be decided. To this end, we will use the following inequality for the Gamma function*

$$\left(x - \frac{1}{2}\right)\ln x - x + \frac{1}{2}\ln(2\pi) < \ln\Gamma(x) < \left(x - \frac{1}{2}\right)\ln x - x + \frac{1}{2}\ln(2\pi) + \frac{1}{12x}, \qquad x > 0. \quad (89)$$

*Taking logarithms,*

$$\ln F(d) = \frac{d}{2}\ln\pi + \ln\Gamma(\lambda d) + \frac{1}{2}\ln\Gamma(d+7/2) - \ln\Gamma((\lambda+1/2)d). \quad (90)$$

*Then using (89), we get*

$$\ln F(d) < \frac{d}{2}\ln\pi + \left(\lambda d - \frac{1}{2}\right)\ln(\lambda d) - \lambda d + \frac{1}{2}(d+3)\ln\left(d + \frac{7}{2}\right)$$

$$- \frac{1}{2}\left(d + \frac{7}{2}\right) - \left(\left(\lambda + \frac{1}{2}\right)d - \frac{1}{2}\right)\ln\left(\left(\lambda + \frac{1}{2}\right)d\right) + \left(\lambda + \frac{1}{2}\right)d \quad (91)$$

$$+ \frac{1}{4}\ln\pi + \frac{1}{12\lambda d} + \frac{1}{12(d+7/2)}.$$

*Upon simplifying, we find that the right-hand side contains a function which is linear in $d$ and one which is sub-linear in $d$, i.e.*

$$\ln F(d) < d\left[\frac{1}{2}\ln\pi + \lambda\ln\lambda - \left(\lambda + \frac{1}{2}\right)\ln\left(\lambda + \frac{1}{2}\right)\right]$$

$$+ d\left[\frac{1}{2}\ln\left(d + \frac{7}{2}\right) - \frac{1}{2}\ln d\right] \quad (92)$$

$$+ \frac{3}{2}\ln(d+7/2) - \frac{1}{2}\ln\lambda d - \lambda - 9/4 - \frac{1}{2}\ln\left(\left(\lambda + \frac{1}{2}\right)d\right).$$

*The second term can be bounded independently of dimension; using the inequality $\ln(1 + x) < x$ for $x > 0$, we have:*

$$d\left[\frac{1}{2}\ln\left(d + \frac{7}{2}\right) - \frac{1}{2}\ln d\right] = \frac{d}{2}\ln\left(1 + \frac{7}{2d}\right) < \frac{d}{2}\left(\frac{7}{2d}\right) = \frac{7}{4}. \quad (93)$$

*Therefore, to prove at most polynomial growth in $d$, it suffices to show that*

$$\mathcal{P}(\lambda, d) := \frac{1}{2} \ln \pi + \lambda \ln \lambda - \left(\lambda + \frac{1}{2}\right) \ln \left(\lambda + \frac{1}{2}\right) \leq 0. \tag{94}$$

*We seek an upper bound for the expression, so we apply the lower bound $\ln(1 + x) > x - \frac{x^2}{2}$ to the logarithm inside the negative term. For $x = \frac{1}{2\lambda}$, this gives us*

$$\ln \left(\lambda + \frac{1}{2}\right) = \ln \lambda + \ln \left(1 + \frac{1}{2\lambda}\right) > \ln \lambda + \frac{1}{2\lambda} - \frac{1}{8\lambda^2}. \tag{95}$$

*Substituting these bounds into the left-hand side of (94), we get*

$$\mathcal{P}(\lambda, d) < \frac{1}{2} \ln \pi + \lambda \ln \lambda - \left(\lambda + \frac{1}{2}\right) \left(\ln \lambda + \frac{1}{2\lambda} - \frac{1}{8\lambda^2}\right)$$

$$< \frac{1}{2} \ln \pi + \lambda \ln \lambda - \left(\lambda \ln \lambda + \frac{1}{2} - \frac{1}{8\lambda} + \frac{1}{2} \ln \lambda + \frac{1}{4\lambda} - \frac{1}{16\lambda^2}\right). \tag{96}$$

*Grouping the constant terms and simplifying, we find:*

$$\mathcal{P}(\lambda, d) < \left(\frac{1}{2} \ln \pi - \frac{1}{2} - \frac{1}{2} \ln \lambda\right) + \left(-\frac{1}{8\lambda} + \frac{1}{16\lambda^2}\right). \tag{97}$$

*For the expression not to grow exponentially as $d \to \infty$, the dominant constant term must be non-positive. Notice that for $\lambda > 1/2$, the expression inside the second pair of brackets is negative. Thus, a sufficient condition is given by:*

$$\frac{1}{2} \ln \pi - \frac{1}{2} - \frac{1}{2} \ln \lambda \leq 0, \tag{98}$$

*i.e.*

$$\lambda \geq \frac{\pi}{e}. \tag{99}$$

*Therefore, choosing $\alpha(d) \geq \frac{\pi}{e} d$ ensures that the constant appearing in front of (14) grows at most polynomially in $d$. Note that we do not expect a better rate than this due to the extra terms appearing in (92).*

### A.6 PROOF OF COROLLARY 2

*Proof.* Since we are dealing with a system of equations, we need to illustrate how the result of Theorem 1 applies. Firstly, given a couple $(\rho, u)$ belonging to class (25), the representation result of Proposition 1 tells us that we can find $\psi_\rho, \psi_u \in \mathcal{S}(\mathbb{R})$ such that we have the representations (assuming $\rho, u$ have been smoothly extended to the full space):

$$\rho(t, x) = \int_{\mathbb{R}} \int_{\mathbb{R}^d} \int_{\mathbb{R}} (A_\rho)(\tau, \mathbf{a}, b) \sigma(\tau t + \mathbf{a} \cdot \mathbf{x} - b) \, d\tau d\mathbf{a} db,$$

$$u(t, x) = \int_{\mathbb{R}} \int_{\mathbb{R}^d} \int_{\mathbb{R}} (A_u)(\tau, \mathbf{a}, b) \sigma(\tau t + \mathbf{a} \cdot \mathbf{x} - b) \, d\tau d\mathbf{a} db, \tag{100}$$

where $A_\rho := \mathcal{R}_{\psi_\rho} \rho$ and $A_u := \mathcal{R}_{\psi_u} u$. Then as in Theorem 1 we can introduce two unbiased estimators

$$\rho_N(t, x) = \frac{1}{N} \sum_{i=1}^{N} \frac{A_\rho(\tau_i, \mathbf{a}_i, b_i)}{\pi(\tau_i, \mathbf{a}_i, b_i)} \sigma(\tau_i t + \mathbf{a}_i \cdot \mathbf{x} - b_i),$$

$$u_N(t, x) = \frac{1}{N} \sum_{i=1}^{N} \frac{A_u(\tau_i, \mathbf{a}_i, b_i)}{\pi(\tau_i, \mathbf{a}_i, b_i)} \sigma(\tau_i t + \mathbf{a}_i \cdot \mathbf{x} - b_i). \tag{101}$$

The result of Theorem 1 gives us the following bounds immediately:

$$\|\rho_N - \rho\|_{L_t^2 H_x^q}^2 \leq \frac{C_\Omega C_\pi \|\sigma^{(q)}\|_\infty^2 T |D| q \mathcal{L}_\psi}{N} \cdot \|\rho\|_{H_t^{s_1} H_x^{q+s_2}}^2,$$

$$\|u_N - u\|_{L_t^2 H_x^q}^2 \leq \frac{C_\Omega C_\pi \|\sigma^{(q)}\|_\infty^2 T |D| q \mathcal{L}_\psi}{N} \cdot \|u\|_{H_t^{s_1} H_x^{q+s_2}}^2, \tag{102}$$

for $s_1, s_2 > 3/2$ since we are in dimension $d = 1$. From now on we label

$$\mathcal{M}_\psi := \frac{C_\Omega C_\pi \|\sigma^{(q)}\|_\infty^2 T|D|q\mathcal{L}_\psi}{N}. \tag{103}$$

The pair $v_N(t, x) := (\rho_N(t, x), u_N(t, x))$ can be interpreted as the outputs of a single random neural network of width $2N$, from which the first claim follows. We now bound the loss functions $\mathcal{J}_{PDE}^1$ and $\mathcal{J}_{PDE}^2$. For $\mathcal{J}_{PDE}^1$ we note the expression $\rho_N u_N - \rho u = \rho_N(u_N - u) + u(\rho_N - \rho)$ to get

$$\partial_x(\rho_N u_N - \rho u) = \partial_x\rho_N(u_N - u) + \rho_N\partial_x(u_N - u) + \partial_x u(\rho_N - \rho) + u\partial_x(\rho_N - \rho). \tag{104}$$

Therefore,

$$\mathcal{J}_{PDE}^1(\mathbf{v}_N) = \int_{(0,T)\times D} |\partial_t(\rho_N - \rho) + \partial_x(\rho_N u_N - \rho u)|^2$$

$$\leq 2\|\partial_t(\rho_N - \rho)\|_{L_{t,x}^2}^2 + 4\|\partial_x\rho_N\|_{L_{t,x}^\infty}^2\|u_N - u\|_{L_{t,x}^2}^2 + 4\|\rho_N\|_{L_{t,x}^\infty}^2\|\partial_x(u_N - u)\|_{L_{t,x}^2}^2$$

$$+ 4\|\partial_x u\|_{L_{t,x}^\infty}^2\|\rho_N - \rho\|_{L_{t,x}^2}^2 + 4\|u\|_{L_{t,x}^\infty}^2\|\partial_x(\rho_N - \rho)\|_{L_{t,x}^2}^2$$

$$\leq 4(\|\partial_x\rho_N\|_{L_{t,x}^\infty}^2 + \|\rho_N\|_{L_{t,x}^\infty}^2 + \|\partial_x u\|_{L_{t,x}^\infty}^2 + \|u\|_{L_{t,x}^\infty}^2)(\|\rho_N - \rho\|_{H_{t,x}^1}^2 + \|u_N - u\|_{H_{t,x}^1}^2)$$

$$=: \mathcal{B}_\mathcal{N}(\|\rho_N - \rho\|_{H_{t,x}^1}^2 + \|u_N - u\|_{H_{t,x}^1}^2). \tag{105}$$

Note that $\|\partial_x u\|_{L_{t,x}^\infty}$ is finite since $u \in C(0, T; H^5(D))$. An upper bound for the coefficient $\mathcal{B}_N$ is $4(\|\rho_N\|_{W_{t,x}^{1,\infty}}^2 + \|u\|_{W_{t,x}^{1,\infty}}^2)$. By a similar Markov argument to the proof of Corollary 1 (see Appendix A.5), for $\delta \in (0, 1)$, we have with probability $1 - \delta$ that

$$\mathcal{J}_{PDE}^1(\mathbf{v}_N) \leq \frac{4(L + \|u\|_{W_{t,x}^{1,\infty}}^2)\mathcal{M}_\psi}{N\delta}(\|\rho\|_{H_t^{1+s_1}H_x^{1+s_2}}^2 + \|u\|_{H_t^{1+s_1}H_x^{1+s_2}}^2), \tag{106}$$

where $s_1, s_2 > 3/2$, if the network weights are sampled such that

$$\|\rho_N\|_{W_{t,x}^{1,\infty}}^2 + \|u\|_{W_{t,x}^{1,\infty}}^2 \leq L < +\infty. \tag{107}$$

For the momentum equation, we use $\rho_N u_N^2 = u_N\rho_N(u_N - u) + uu_N(\rho_N - \rho)$, and so

$$\partial_x(\rho_N u_N^2) - \partial_x(\rho u^2) = \partial_x(u_N\rho_N)(u_N - u) + u_N\rho_N\partial_x(u_N - u) + \partial_x(uu_N)(\rho_N - \rho)$$

$$+ uu_N\partial_x(\rho_N - \rho). \tag{108}$$

Therefore, in a similar way to the continuity equation, with probability $1 - \delta$,

$$\mathcal{J}_{PDE}^2(\mathbf{v}_N) = \int_{(0,T)\times D} |\partial_t(\rho_N u_n - \rho u) + \partial_x(\rho_N u_N^2 - \rho u^2) - \mu\partial_x^2(u_N - u)|^2 dxdt$$

$$\leq \mathcal{C}_N\frac{\mathcal{M}_\psi}{N\delta}(\|\rho_N - \rho\|_{H_{t,x}^1}^2 + (\mu + 1)\|u_N - u\|_{H_t^1 H_x^2}^2), \tag{109}$$

where

$$\mathcal{C}_N = \|\partial_t\rho_N\|_{L_{t,x}^\infty}^2 + \|\rho_N\|_{L_{t,x}^\infty}^2 + \|\partial_t u\|_{L_{t,x}^\infty}^2 + \|u\|_{L_{t,x}^\infty}^2 + \|\partial_x(\rho_N u_N)\|_{L_{t,x}^\infty}^2$$

$$+ \|\rho_N u_N\|_{L_{t,x}^\infty}^2 + \|\partial_x(uu_N)\|_{L_{t,x}^\infty}^2 + \|uu_N\|_{L_{t,x}^\infty}^2. \tag{110}$$

The assumption (107) ensures that $\mathcal{C}_N$ is bounded by

$$2(L^2 + L + L\|u\|_{W_{t,x}^{1,\infty}}^2 + \|u\|_{W_{t,x}^{1,\infty}}^2) \leq 2(L + 1)(L + \|u\|_{W_{t,x}^{1,\infty}}^2). \tag{111}$$

Note that $\|\partial_t u\|_{L_{t,x}^\infty}$ is finite; this can be seen by rewriting the momentum equation as $\partial_t u = -u\partial_x u + \mu\rho^{-1}\partial_x^2 u$, and using the regularity $u \in C(0, T; H^k(D))$ for $k \geq 5$, along with the Sobolev embedding $H^1 \hookrightarrow L^\infty$. Then we have with probability $1 - \delta$ that

$$\mathcal{J}_{PDE}^2(\mathbf{v}_N) \leq \frac{2(L + 1)(L + \|u\|_{W_{t,x}^{1,\infty}}^2)\mathcal{M}_\psi}{N\delta}(\|\rho\|_{H_t^{1+s_1}H_x^{1+s_2}}^2 + (\mu + 1)\|u\|_{H_t^{1+s_1}H_x^{2+s_2}}^2).$$

Noting that $s_1, s_2 > 3/2$, we get

$$(\mathcal{J}_{PDE}^1 + \mathcal{J}_{PDE}^2)(\mathbf{v}_N) \leq \frac{2(L+1)(L + \|u\|_{W_{t,x}^{1,\infty}}^2)\mathcal{M}_\psi}{N\delta}(\|\rho\|_{H_t^3 H_x^3}^2 + (\mu+1)\|u\|_{H_t^3 H_x^4}^2), \quad (112)$$

which is guaranteed to be finite if we take $(\rho_0, u_0) \in (H^k(D))^2$ for $k \geq 5$. This can be seen by noting that the continuity equation can be re-written as $\partial_t \rho = -u\partial_x \rho - \rho\partial_x u$, and the momentum equation as $\partial_t u = -u\partial_x u + \mu\rho^{-1}\partial_x^2 u$. Then using the regularity (25), one can bound the higher order Sobolev norms on the right-hand side of (112). $\qquad\square$

## B    EXPERIMENTS

In this section we give further details on the numerical experiments carried out in Section 5. All experiments were performed using a NVIDIA RTX 3080 GPU.

### B.1    POROUS MEDIUM EQUATION

Recall that we consider the Barenblatt-Kompaneets-Zeldovich solution

$$u(t,x) = \frac{1}{t^\alpha}\left(b - \frac{m-1}{2m}\beta\frac{\|x\|^2}{t^{2\beta}}\right)_+^{\frac{1}{m-1}}, \quad (113)$$

where $\|\cdot\|$ is the $\ell^2$ norm, $(\cdot)_+$ is the positive part and $\alpha = \frac{d}{d(m-1)+2}$ and $\beta = \frac{1}{d(m-1)+2}$. We set $m = 2$.

This solution is compactly supported but not differentiable at the edges of the support, which causes difficulty for numerical schemes. Since this solution is generated by a Dirac delta-valued initial data, we fix a small time $t_0 = 10^{-2}$ and take the initial data to be $u_0(x) := u(t_0, x)$ and solve on the shifted time domain $(t_0, T + t_0)$.

We perform two types of experiments. First, we investigate the effect of the width $N$ of the network on the relative error between the network and the solution $u$, and then we compare peformance of the randomised architecture against traditional PINN architectures in dimensions $d = 1 - 5$.

#### B.1.1    EFFECT OF NETWORK WIDTH ON THE ERROR

We aim to investigate whether a convergence rate of $N^{-1/2}$ can be observed in practice. To this end, we train a RaNN to approximate the Barenblatt profile (29) in dimensions $d = 1, ..., 5$. In each dimension, we take a set of widths $N \in \{N_1, ..., N_k\}$, train the network for each width and plot the relative error of the final network against the true solution. The RaNN includes a Fourier feature layer, where frequencies $\omega_j$ are sampled from $\mathcal{N}(0, 10^2)$. For each dimension $d$ and width $N$, we sample $M = 10N$ points (to ensure the problem remains well-posed) with a mixed strategy: $50\%$ of the points are sampled uniformly and $50\%$ are sampled uniformly on $[0.2, 0.8]^d$, which is a box focused on the support of the solution. Then we find weights $\mathbf{W} = \{W_i\}_{i=1}^N$ which minimise the Ridge regression loss

$$\mathcal{L}(\mathbf{W}) = \frac{1}{M}\sum_{i=1}^M \|\hat{\mathbf{u}}(t_i, \mathbf{x}_i) - \mathbf{y}_i\|_2^2 + \lambda\|\mathbf{W}\|_2^2, \qquad \lambda = 10^{-5}. \quad (114)$$

We compute the closed-form solution $\hat{\mathbf{W}}$ directly (using a Cholesky decomposition), and then evaluate the relative error. We train the RaNN five times for each width. The mean relative $L^2$ errors are plotted against the widths in Figure 3. The errors are also plotted on log-log scales in Figure 4. The key observation here is that the RaNN error points lie close to the $C/\sqrt{N}$ curve, which supports the upper bound of Theorem 1. Note that in $d = 1, 2, 3$ we take $t_0 = 0.1$ (i.e. the initial data is set to $u(t_0, x)$, where $u$ is the Barenblatt profile from (29)) and in $d = 4, 5$ this is reduced to $t_0 = 0.01$ to keep the problem computationally manageable.

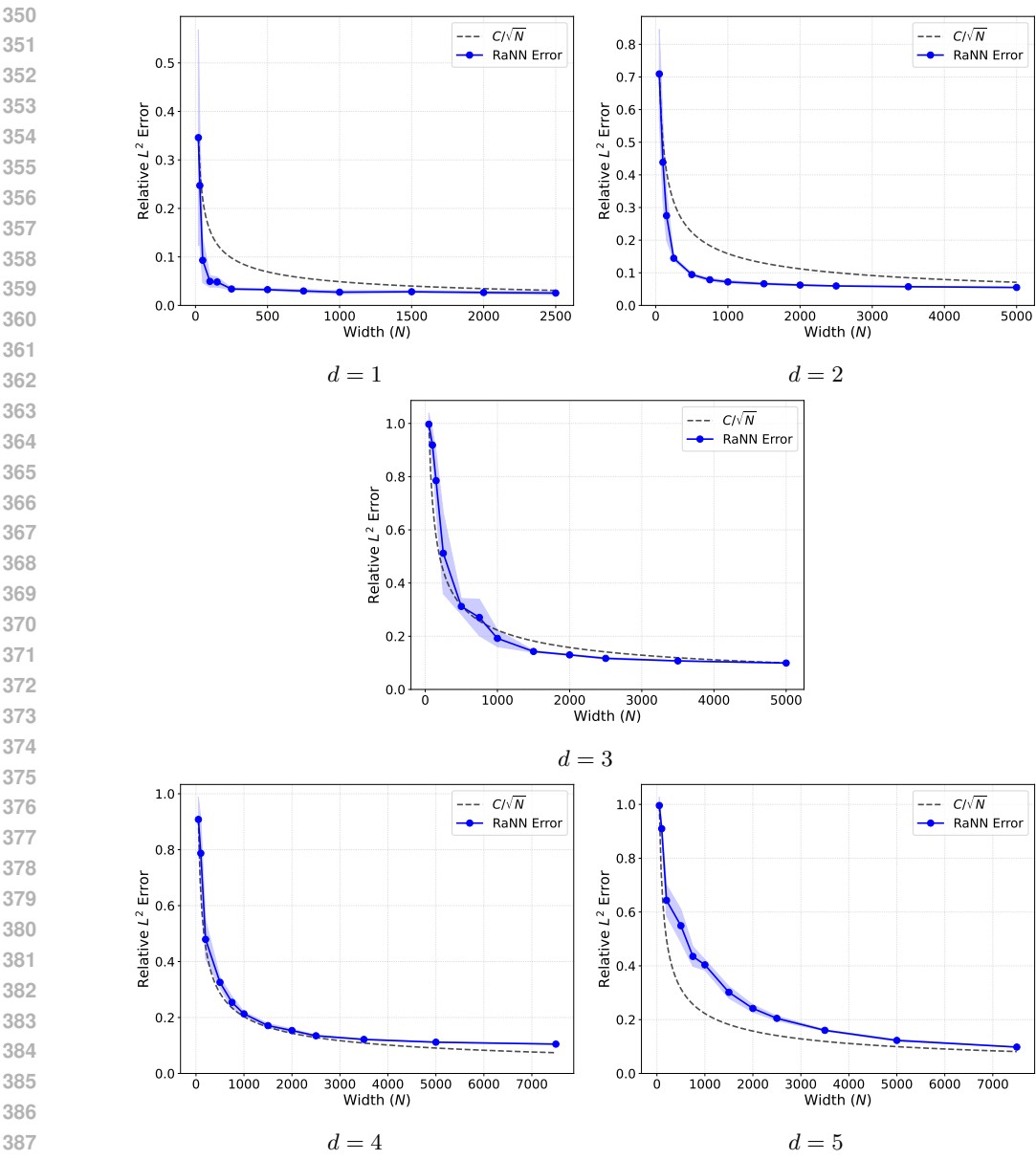

Figure 3: Approximation error of RaNNs of varying width for solving PMEs in dimensions $d = 1, \ldots, 5$. The shaded band indicates the region within one standard deviation of the mean relative $L^2$ error.

### B.1.2 COMPARISON WITH PINNs

Here, we provide full experimental details and detailed results for the PME experiments from Section 5.1.2.

**Architectures** For our main experiments for the PME we consider three architectures:

- **RaNN:** a random neural network in the sense of Definition 1, which includes a Fourier feature embedding layer; this technique has been shown to enhance the performance of physics-informed solvers (see Tancik et al. (2020)). A more detailed description of Fourier features is given in Appendix C. We use $N = 2500, 5000, 7500$ features in $d = 1, 2, 3$

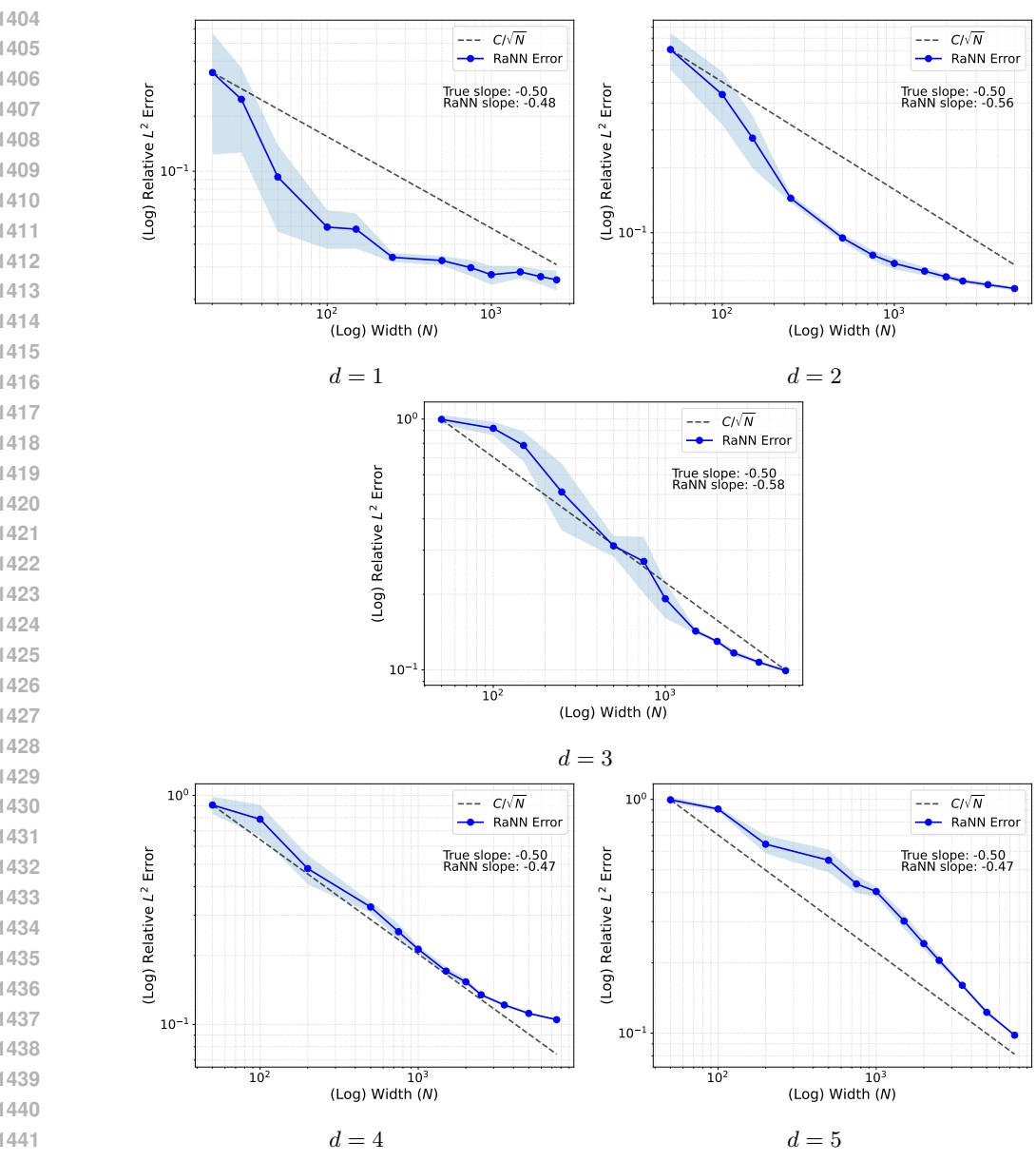

Figure 4: Log-log plot of the relative $L^2$ error versus the width $N$. The reference scaling $C/\sqrt{N}$ and the measured RaNN slope are shown. The shaded band indicates the region within one standard deviation of the mean relative $L^2$ error.

respectively and $N = 7500$ for $d = 4, 5$ in order to maintain computational feasibility. We sample $(\tau_i, \mathbf{a}_i) \sim \mathcal{N}(0, 10^2 \mathbb{I}_{d+1})$, $b_i \sim \text{Unif}[0, 2\pi]$. Since we include a Fourier embedding layer, the parameters $(\tau_i, \mathbf{a}_i)$ can be interpreted the frequency of the $\sin / \cos$ features (see the construction of (122)). In our tests we find that using higher frequency features helps to learn solutions in higher dimensions, which is why we choose the large variance of $10^2 \mathbb{I}_{d+1}$. We find that this choice of distribution leads to slightly more reliable training results, compared to distributions such as (12), which was used in Theorem 1.

- **PINN (A):** A physics-informed neural network which mimics the architecture of RaNN for each dimension $d$, but where all parameters are trainable. This network therefore has many

more trainable parameters than the RaNN. Weights are initialised using a Xavier (Glorot) scheme.

- **PINN (B):** A more traditional physics-informed neural network with four hidden layers containing $N = 20, 30, 40, 48, 54$ nodes in dimensions $d = 1, 2, 3, 4, 5$ respectively. The $\tanh$ activation is used. The widths $N$ are chosen so that the total number of trainable parameters is roughly equal to that of the RaNN, for any given dimension. Weights are initialised using a Xavier (Glorot) scheme.

**Loss** For each architecture, the loss function used is $\mathcal{L} := \mathcal{L}_{PDE} + \mathcal{L}_{IC} + \mathcal{L}_{BC}$, where

$$\mathcal{L}_{PDE}(u_\theta) = \lambda_{PDE} \frac{1}{M_p} \sum_{i=1}^{M_p} |\partial_t u_\theta - \Delta(u_\theta^2)|^2 (t_i^p, x_i^p),$$

$$\mathcal{L}_{BC}(u_\theta) = \lambda_{BC} \left[ \frac{1}{M_b} \sum_{i=1}^{M_b} |u_\theta(t_i^b, 1)|^2 + \frac{1}{M_{bc}} \sum_{i=1}^{M_b} |u_\theta(t_i^b, 0)|^2 \right], \qquad (115)$$

$$\mathcal{L}_{IC}(u_\theta) = \lambda_{IC} \frac{1}{M_c} \sum_{i=1}^{M_c} |u_\theta(0, x_i^c) - u(x_i^c)|^2,$$

where $\{(t_i^p, x_i^p)\}_{i=1}^{M_p}, \{(t_i^b)\}_{i=1}^{M_p}, \{(x_i^c)\}_{i=1}^{M_p}$ are the points sampled for evaluating the PDE, BC and IC losses respectively. For $d = 1$ we use $T = 0.05$, $M_p = 2000$ collocation points to minimise the PDE residual and $M_c = 1000, M_b = 500$ points for the initial and boundary residuals respectively. For $d = 2$ we choose $T = 0.025$ and $M_p = 4000, M_c = 2000, M_b = 1000$ points. For $d = 3$ we choose $T = 0.01$ and use $M_p = 6000, M_c = 4000, M_b = 2000$ points. For $d = 4, 5$ we stick with $T = 0.01$ and use $M_p = 8000, M_c = 4000, M_b = 2000$ points. Note that we choose the final time $T$ small enough so that the Barenblatt profile is compactly supported on $(0, 1)^d$ for each time, which is reflected in the formulation of $\mathcal{L}_{BC}$. We choose $\lambda_{PDE} = \lambda_{BC} = 1$ and pick a higher weight $\lambda_{IC} = 200$, to ensure the initial condition is satisfied. In $d = 4, 5$ this is increased to $\lambda_{IC} = 400$. We find that in practice this yields smaller overall errors than using uniform weights. Also, since the target solution is non-negative in this case, the final prediction is chosen as the square of the neural network output, i.e. $u_{pred}(t, x) = (u_\theta(t, x))^2$.

**Training** For training, we use the Adam optimiser for the first 4750 epochs and L-BFGS for the remaining 250 epochs. This split is chosen to balance accuracy and computational feasibility. We use the Xavier (Glorot) initialisation for PINN (A) and PINN (B), while the output weights for the RaNN are initialised from a uniform distribution. We use the learning rate $\mu = 10^{-3}$. Simulations were performed using a NVIDIA RTX 3080 GPU.

**Results** The results are shown in Table 2. The RaNN is trained considerably faster than the alternative PINNs, while maintaining a similar degree of accuracy. These results support the viewpoint that randomised architectures are highly capable of learning complex functions, despite their simple form. For $d = 1$ we also include illustrative solution plots in Figures 7 and 8. In each section (dimension) of Table 2, the first row $\mathcal{X}$ corresponds to the $L^2$ average relative error of the network over space-time compared to the true solution whereas the second row $\mathcal{X}_T$ tracks only the error at the final time $T$, i.e.

$$\mathcal{X} := \frac{\|u - u_N^W\|_{L_{t,x}^2}}{\|u\|_{L_{t,x}^2}}, \quad \mathcal{X}_T := \frac{\|u(T) - u_N^W(T)\|_{L_x^2}}{\|u(T)\|_{L_x^2}}.$$

$\sigma_\mathcal{X}$ is then the standard deviation of the space-time relative error for all 5 runs, whereas $\sigma_{\mathcal{X}_T}$ is the standard deviation for the final time relative error.

### B.1.3 ADDITIONAL EXPERIMENTS

In this section, we report additional experimental results for $d = 1, 2, 3$, complementing those reported in Section 5.1.2 and Appendix B.1.2.

| $d$ | Metric | PINN (A) | PINN (B) | RaNN |
|---|---|---|---|---|
| 1 | $\mathcal{X} := \|u - u_N^W\|_{L^2_{t,x}}/\|u\|_{L^2_{t,x}}$ | $7.09 \times 10^{-2}$ | $6.88 \times 10^{-2}$ | $6.41 \times 10^{-2}$ |
| | $\mathcal{X}_T := \|u(T) - u_N^W(T)\|_{L^2_x}/\|u(T)\|_{L^2_x}$ | $4.62 \times 10^{-2}$ | $7.85 \times 10^{-2}$ | $4.62 \times 10^{-2}$ |
| | $\sigma_{\mathcal{X}}$ | $1.38 \times 10^{-3}$ | $2.80 \times 10^{-2}$ | $3.98 \times 10^{-3}$ |
| | $\sigma_{\mathcal{X}_T}$ | $5.58 \times 10^{-3}$ | $3.76 \times 10^{-2}$ | $7.87 \times 10^{-3}$ |
| | Time (mean) | 101s | 164s | 68s |
| | # Trainable params. | 6251 | 2665 | 2500 |
| 2 | $\mathcal{X}$ | $1.08 \times 10^{-1}$ | $1.25 \times 10^{-1}$ | $1.00 \times 10^{-1}$ |
| | $\mathcal{X}_T$ | $1.11 \times 10^{-1}$ | $1.33 \times 10^{-1}$ | $1.06 \times 10^{-1}$ |
| | $\sigma_{\mathcal{X}}$ | $2.38 \times 10^{-3}$ | $8.43 \times 10^{-3}$ | $2.32 \times 10^{-3}$ |
| | $\sigma_{\mathcal{X}_T}$ | $3.71 \times 10^{-3}$ | $6.24 \times 10^{-3}$ | $3.38 \times 10^{-3}$ |
| | Time (mean) | 208s | 152s | 86s |
| | # Trainable params. | 15001 | 4899 | 5000 |
| 3 | $\mathcal{X}$ | $1.24 \times 10^{-1}$ | $2.83 \times 10^{-1}$ | $1.18 \times 10^{-1}$ |
| | $\mathcal{X}_T$ | $1.44 \times 10^{-1}$ | $3.04 \times 10^{-1}$ | $1.48 \times 10^{-1}$ |
| | $\sigma_{\mathcal{X}}$ | $2.90 \times 10^{-3}$ | $1.48 \times 10^{-1}$ | $1.39 \times 10^{-3}$ |
| | $\sigma_{\mathcal{X}_T}$ | $8.17 \times 10^{-3}$ | $1.65 \times 10^{-1}$ | $1.59 \times 10^{-2}$ |
| | Time (mean) | 579s | 187s | 107s |
| | # Trainable params. | 26251 | 7451 | 7500 |
| 4 | $\mathcal{X}$ | $1.60 \times 10^{-1}$ | $3.48 \times 10^{-1}$ | $1.68 \times 10^{-1}$ |
| | $\mathcal{X}_T$ | $2.69 \times 10^{-1}$ | $4.16 \times 10^{-1}$ | $2.97 \times 10^{-1}$ |
| | $\sigma_{\mathcal{X}}$ | $5.35 \times 10^{-3}$ | $1.51 \times 10^{-1}$ | $1.67 \times 10^{-2}$ |
| | $\sigma_{\mathcal{X}_T}$ | $8.49 \times 10^{-3}$ | $1.50 \times 10^{-1}$ | $3.56 \times 10^{-2}$ |
| | Time (mean) | 1034s | 225s | 138s |
| | # Trainable params. | 30001 | 8189 | 7500 |
| 5 | $\mathcal{X}$ | $5.07 \times 10^{-1}$ | $5.33 \times 10^{-1}$ | $3.78 \times 10^{-1}$ |
| | $\mathcal{X}_T$ | $6.75 \times 10^{-1}$ | $6.84 \times 10^{-1}$ | $3.96 \times 10^{-1}$ |
| | $\sigma_{\mathcal{X}}$ | $1.80 \times 10^{-1}$ | $1.94 \times 10^{-1}$ | $2.45 \times 10^{-3}$ |
| | $\sigma_{\mathcal{X}_T}$ | $2.27 \times 10^{-1}$ | $1.89 \times 10^{-1}$ | $2.81 \times 10^{-2}$ |
| | Time (mean) | 1395s | 326s | 168s |
| | # Trainable params. | 33751 | 10024 | 7500 |

Table 2: Error norms and computational time for varying $d$ and $N$ values.

The experiment here includes an additional network: a randomised neural network without Fourier feature layer. Moreover, for this experiment we chose the network sizes so that comparable errors were achieved between PINNs and RaNNs in order to focus on comparing training times.

**Architectures** We train the following networks:

- **RaNN (A):** a random neural network in the sense of Definition 1, where we randomly sample $(\tau_i, \mathbf{a}_i) \sim \mathcal{N}(0, 10^2 \mathbb{I}_{d+1})$, $b_i \sim \text{Unif}[0, 2\pi]$. We also include a Fourier feature embedding layer. A more detailed description of Fourier features is given in Appendix C. We use $N = 2500, 5000, 7500$ features in $d = 1, 2, 3$ respectively. Weights are initialised from a uniform distribution.

- **RaNN (B):** a random neural network in the sense of Definition 1, where we randomly sample $(\tau_i, \mathbf{a}_i) \sim \mathcal{N}(0, 10^2 \mathbb{I}_{d+1})$, $b_i \sim \text{Unif}[0, 2\pi]$. Weights are initialised from a uniform distribution.

- **PINN (A):** A classical physics-informed neural network with one Fourier feature layer (containing $N = 2500, 5000, 7500$ features in $d = 1, 2, 3$ respectively) and no other hidden layers. The `tanh` activation is used. Weights are initialised using Xavier (Glorot) initialisation.

- **PINN (B):** A classical physics-informed neural network with two hidden layers containing $N = 100, 125, 150$ nodes in dimensions $d = 1, 2, 3$ respectively. The `tanh` activation is used. Weights are initialised using Xavier (Glorot) initialisation.

**Loss** We use the same setup and loss function (115) as in Appendix B.1.2. In particular, the choices of $\lambda_{PDE} = \lambda_{BC} = 1$ and $\lambda_{IC} = 200$ are as before. For $d = 1$ we use $T = 0.05$, $M_p = 2500$ collocation points to minimise the PDE residual and $M_c = 1250$, $M_b = 625$ points for the initial and boundary residuals respectively. For $d = 2$ we choose $T = 0.025$ and $M_p = 5000$, $M_c = 2500$, $M_b = 1250$ points. Finally, for $d = 3$ we choose $T = 0.01$ and use $M_p = 7500$, $M_c = 5000$, $M_b = 2500$ points. As before, since the target solution is non-negative in this case, the final prediction is chosen as the square of the neural network output, i.e. $u_{pred}(t, x) = (u_\theta(t, x))^2$.

**Training** As ablation to the previous experiment, we use an Adam optimiser for 5000 epochs. As before, we use the Xavier (Glorot) initialisation for PINN (A) and PINN (B), while the output weights for the RaNN are initialised from a uniform distribution. We use the learning rate $\mu = 10^{-3}$. Simulations were performed using a NVIDIA RTX 3080 GPU.

**Results** The results are given in Table 3. Similar to our main experiments, the randomised network RaNN (A) performs faster than both PINNs and with a similar magnitude of error. A key observation here is that RaNN (B) without Fourier features, although fastest, suffers from poor accuracy. This suggests that Fourier features help boost the expressivity of the network with minimal additional computational cost (only $12s$ longer training time in $d = 2$). Another interesting point is that RaNN

| $d$ | Metric | PINN (A) | PINN (B) | RaNN (A) | RaNN (B) |
|---|---|---|---|---|---|
| 1 | $\mathcal{X}$ | $5.80 \times 10^{-2}$ | $5.86 \times 10^{-2}$ | $5.79 \times 10^{-2}$ | $9.94 \times 10^{-1}$ |
| | $\mathcal{X}_T$ | $4.57 \times 10^{-2}$ | $5.12 \times 10^{-2}$ | $4.56 \times 10^{-2}$ | $9.92 \times 10^{-1}$ |
| | $\sigma_{\mathcal{X}}$ | $2.18 \times 10^{-3}$ | $7.49 \times 10^{-3}$ | $2.34 \times 10^{-3}$ | $7.28 \times 10^{-3}$ |
| | $\sigma_{\mathcal{X}_T}$ | $2.73 \times 10^{-3}$ | $7.53 \times 10^{-3}$ | $2.16 \times 10^{-3}$ | $8.72 \times 10^{-3}$ |
| | Time (mean) | 90s | 81s | 50s | 43s |
| | # Trainable params. | 6251 | 10501 | 2500 | 2500 |
| 2 | $\mathcal{X}$ | $1.06 \times 10^{-1}$ | $1.71 \times 10^{-1}$ | $1.06 \times 10^{-1}$ | $2.89 \times 10^{0}$ |
| | $\mathcal{X}_T$ | $8.85 \times 10^{-2}$ | $1.50 \times 10^{-1}$ | $9.01 \times 10^{-2}$ | $3.17 \times 10^{0}$ |
| | $\sigma_{\mathcal{X}}$ | $3.73 \times 10^{-3}$ | $6.92 \times 10^{-3}$ | $3.56 \times 10^{-3}$ | $6.95 \times 10^{-1}$ |
| | $\sigma_{\mathcal{X}_T}$ | $2.59 \times 10^{-3}$ | $6.80 \times 10^{-3}$ | $3.98 \times 10^{-3}$ | $9.71 \times 10^{-1}$ |
| | Time (mean) | 217s | 109s | 67s | 55s |
| | # Trainable params. | 12501 | 16376 | 5000 | 5000 |
| 3 | $\mathcal{X}$ | $1.40 \times 10^{-1}$ | $3.62 \times 10^{-1}$ | $1.44 \times 10^{-1}$ | $1.07 \times 10^{1}$ |
| | $\mathcal{X}_T$ | $1.23 \times 10^{-1}$ | $3.32 \times 10^{-1}$ | $1.27 \times 10^{-1}$ | $1.17 \times 10^{1}$ |
| | $\sigma_{\mathcal{X}}$ | $3.85 \times 10^{-3}$ | $1.74 \times 10^{-2}$ | $5.98 \times 10^{-3}$ | $4.86 \times 10^{0}$ |
| | $\sigma_{\mathcal{X}_T}$ | $7.86 \times 10^{-3}$ | $1.82 \times 10^{-2}$ | $3.37 \times 10^{-3}$ | $5.36 \times 10^{0}$ |
| | Time (mean) | 666s | 142s | 95s | 62s |
| | # Trainable params. | 18751 | 23251 | 7500 | 7500 |

Table 3: Error norms and computational time for varying $d$ and $N$ values.

(A) also uses fewer trainable parameters than the PINNs. Note that in each section (dimension) of Table 3, the first row $\mathcal{X}$ corresponds to the $L^2$ average relative error of the network over space-time compared to the true solution whereas the second row $\mathcal{X}_T$ tracks only the error at the final time $T$,

i.e.

$$\mathcal{X} := \frac{\|u - u_N^W\|_{L_{t,x}^2}}{\|u\|_{L_{t,x}^2}}, \quad \mathcal{X}_T := \frac{\|u(T) - u_N^W(T)\|_{L_x^2}}{\|u(T)\|_{L_x^2}}.$$

$\sigma_{\mathcal{X}}$ is the standard deviation of the space-time relative error for all 5 runs whereas $\sigma_{\mathcal{X}_T}$ is the standard deviation for the final time relative error.

### B.2 NAVIER-STOKES

We now turn to the compressible Navier-Stokes system, given by (22). We restrict our attention to the one-dimensional case, where (22) is reduced to (in Lagrangian mass coordinates)

$$\begin{cases} \partial_t v - \partial_x u = 0, \\ \partial_t u + \partial_x p_\epsilon(v) - \mu \partial_x \left( \frac{1}{v} \partial_x u \right) = 0. \end{cases} \tag{116}$$

Here, $v = 1/\rho$ represents the specific volume and $u$ the velocity. We consider the singular pressure in the work of Dalibard & Perrin (2020).

$$p_\epsilon(v) = \frac{\epsilon}{(v-1)^\gamma}, \qquad \gamma > 0. \tag{117}$$

In order to evaluate the performance of the randomised PINN method, we need a baseline solution analogous to the Barenblatt profile ((29)) which we used for the Porous Medium Equation. For this purpose, we consider the travelling shock-wave solutions to system (116) for this system, which were studied by Dalibard & Perrin (2020). Shock wave solutions to compressible Navier-Stokes models have also been studied in other works (Mascia & Zumbrun (2004); Dalibard & Perrin (2024); Humpherys et al. (2010); Vasseur & Yao (2016)). The travelling wave solutions to system (116) can be obtained by taking the ansatz $(v, u)(t, x) = (\mathfrak{v}, \mathfrak{u})(x - st)$, where $s$ is the shock speed. This reduces the PDE to an ODE for $\mathfrak{v}$:

$$\mathfrak{v}' = \frac{\mathfrak{v}}{\mu s}(s^2(v_- - \mathfrak{v}) + p_\epsilon(v_-) - p_\epsilon(\mathfrak{v})). \tag{118}$$

The velocity $\mathfrak{u}$ can then be obtained from the relationship $\mathfrak{v} = -s\mathfrak{u}$ which follows from the conservation of mass.

We consider the domain $(0, T) \times (-5, 5)$ with $T = 1.0, \mu = 1, \epsilon = 10^{-3}, \gamma = 2$. The shock profile connects a far-field state $v_-$ to the far-field state $v_+$. We fix $v_+ = 1.5$, while $v_- < v_+$ and the shock speed $s$ are derived from the Rankine-Hugoniot jump condition (see Proposition 1.1 of Dalibard & Perrin (2020)).

To obtain the baseline solution $(\mathfrak{v}, \mathfrak{u})$, we numerically integrate the ODE using the scipy.integrate.odeint solver on an interval $\xi \in [-5, 5]$ with 5000 points. The velocity profile $\mathfrak{u}(\xi)$ is then obtained from $\mathfrak{v} = -s\mathfrak{u}$. The travelling wave solution can be seen below in Figure 5.

In an effort to minimise optimisation error and adhere to the setting of Theorem 1 as closely as possible, we choose to use a supervised learning approach when finding a Randomised Neural Network. We include Fourier features (see Appendix C for details) where frequencies $\omega_j$ are drawn from $\mathcal{N}(0, 3.5^2)$. Note that we choose to use a smaller variance than for the PME because the solution is of lower frequency than the Barenblatt profile.

We sweep across a range of widths $N \in \{10, ..., 250\}$ and aim to minimise the mean-squared error between the network and the solution to the ODE. For each width, we train the model using Ridge regression on a dataset where the sample size is $M = 2000N$ ($M$ grows with $N$ to avoid an underdetermined problem, ensuring the optimisation problem remains stable). For any given training point $(t_i, x_i)$, the evaluation of the baseline solution $(\mathfrak{v}, \mathfrak{u})$ is obtained by linear interpolation on the ODE grid. We sample uniformly in time but use a mixture of uniform and importance sampling in space; 50% of points are sampled uniformly on $[-5, 5]$ whereas 50% of points are sampled from a normal distribution $N(x_0, 1)$ around the shock location $x_0(t) = x_0 - st$. The frequencies for the Fourier features $\omega_j$ are sampled from $\mathcal{N}(0, 3.5^2)$. We find the smaller variance of $3.5^2$ to be effective for the simpler behaviour of a travelling wave solution.

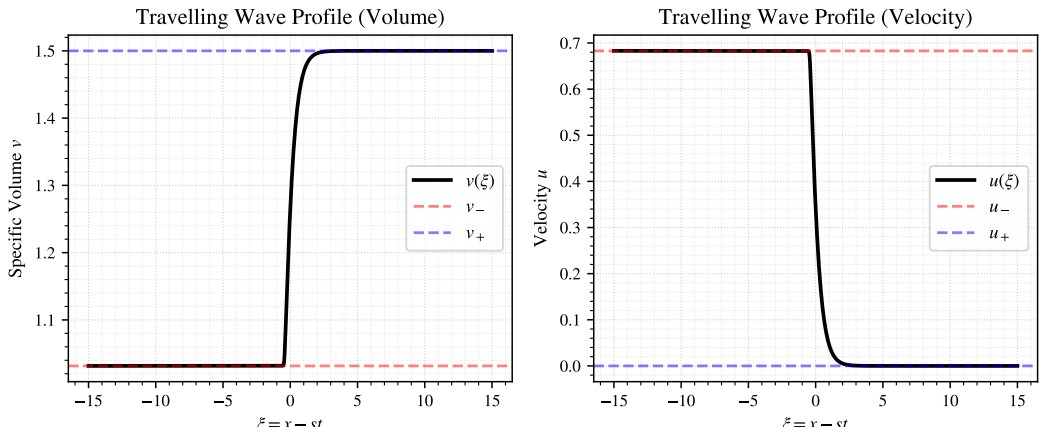

Figure 5: The travelling wave solution $(\mathfrak{v}, \mathfrak{u})$ to (116), obtained by solving the ODE (118).

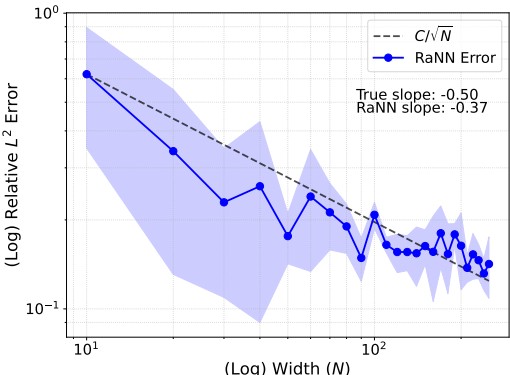

Figure 6: Approximation error of RaNNs of varying width $N$ for solving the compressible Navier-Stokes system on $(0, 1) \times [-5, 5]$ in logarithmic scale.

The network is trained to minimise the $L^2$-regularised MSE (Ridge regression loss) :

$$\mathcal{L}(\mathbf{W}) = \frac{1}{M} \sum_{i=1}^{M} \|\hat{\mathbf{v}}(t_i, \mathbf{x}_i) - \mathbf{y}_i\|_2^2 + \lambda \|\mathbf{W}\|_2^2, \qquad \lambda = 10^{-3}, \tag{119}$$

where $\hat{\mathbf{v}}$ is the network output and $\mathbf{W}$ is the vector of output weights. The minimiser $\hat{\mathbf{W}}$ has a closed-form which can be explicitly calculated and used to generate the final network. With the final network, we compute the $L^2$ errors relative to the baseline solution using a set of $20,000$ (pre-generated) points. Each width $N$ is tested five times and the mean relative error is recorded for each $N$. These errors are plotted against $N$ in Figure 2. The errors are plotted on a log-log scale in Figure 6.

## C  RANDOM FEATURE NEURAL NETWORKS

It is known that classical PINNs suffer from a spectral bias phenomenon Rahaman et al. (2019), which essentially means that the network is biased towards learning lower frequency functions (seeWang et al. (2022; 2023); Xu et al. (2019)). This can be troublesome, particularly for nonlinear PDEs whose solutions are often highly chaotic. Tancik et al. (2020) suggested to use Fourier features to overcome this.

To carry out the simulations in Section 4, we integrated Fourier feature embeddings into the RaNN network. We now describe the architecture of a network with Fourier feature embeddings. Instead

of choosing a smooth activation such as $\tanh$, we take $\sigma(z) = \cos(z)$ and include both cosine and sine activations for symmetry. Concretely, let $\{\tau_i, \mathbf{a}_i\}_{i=1}^N \subset \mathbb{R}^{1+d}$ and $\{b_i\}_{i=1}^N \subset [0, 2\pi]$ be frozen random samples, and define

$$\phi_i(t, x) = \cos(\tau_i t + \mathbf{a}_i \cdot x + b_i), \qquad \psi_i(t, x) = \sin(\tau_i t + \mathbf{a}_i \cdot x + b_i). \tag{120}$$

We then construct the feature vector

$$\Phi(t, x) = \frac{1}{\sqrt{N}} \big( \phi_1(t, x), \ldots, \phi_N(t, x), \psi_1(t, x), \ldots, \psi_N(t, x) \big), \tag{121}$$

and take

$$u_W(t, x) = \beta + \sum_{i=1}^N \big( a_i \phi_i(t, x) + c_i \psi_i(t, x) \big), \tag{122}$$

where the coefficients $\{a_i, c_i\}_{i=1}^N$ and bias $\beta$ are the trainable parameters. The prefactor $N^{-1/2}$ normalises the variance of the features, and does not affect the approximation class.

We can express the sum of sine and cosine functions as a single shifted cosine with amplitude $W_i$, giving us the form

$$u_W(t, x) = \beta + \sum_{i=1}^N W_i \cos(\tau_i t + \mathbf{a}_i \cdot \mathbf{x}_i + \tilde{b}_i), \tag{123}$$

where $\tilde{b}_i := b_i - \theta_i$. This shows that the RaNN used in our experiments is of the same general form as (1), with smooth activation $\sigma = \cos$. Recall from Remark 3 that $\sigma = \cos$ is an admissible choice, meaning that the approximation result of Theorem 1 directly applies to networks of the form (122).

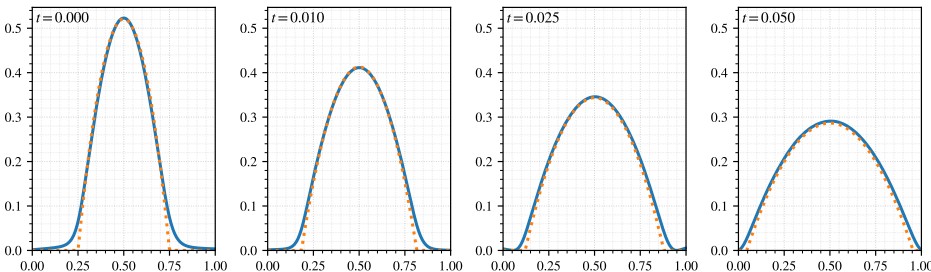

Figure 7: PINN (A) solution (5000 epochs, $T = 0.05$, 1250 Fourier features) given by the blue curve. The true Barenblatt solution is given by the orange dotted curve.

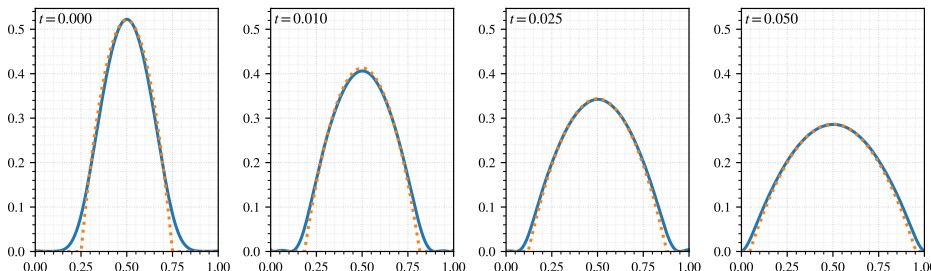

Figure 8: RaNN (A) solution (5000 epochs, $T = 0.05$, 1250 Fourier features) given by the blue curve. The true Barenblatt solution is given by the orange dotted curve.

