# OpenReview forum: "Random Neural Network Expressivity for Non-Linear Partial Differential Equations"
_ICLR.cc/2026/Conference — Submitted to ICLR 2026_

### Official Review · Reviewer_TgGQ · 2025-10-20

**Soundness:** 2
**Presentation:** 1
**Contribution:** 3
**Rating:** 2
**Confidence:** 4

**Summary:**

This paper investigates the expressivity and approximation capabilities of Random Neural Networks (RaNNs) for solving non-linear, time-dependent partial differential equations (PDEs). The authors aim to provide theoretical guarantees for RaNNs in the context of Physics-Informed Neural Networks (PINNs), showing that such models with randomly generated weights can achieve dimension-independent approximation rates under Sobolev norms.

**Strengths:**

- The paper establishes a theoretical result (Theorem 1) showing that RaNNs can approximate time-dependent Sobolev functions at a rate of $N^{-1/2}$, where $N$ is the number of random neurons.
Importantly, this bound is independent of the spatial dimension, suggesting that RaNNs can mitigate the curse of dimensionality in high-dimensional PDE problems.
- De Ryck et al. (2025) required strong smoothness assumptions, $ s \geq (d+9)/2$, to estimate $H^1$ and $H^2$ errors. This paper relaxes these regularity requirements substantially.
- Previous RaNN approximation results (including De Ryck et al., 2025; Neufeld & Schmocker, 2023) mainly addressed static function approximation, either spatial functions or solutions at a fixed time. This paper extends the theory to time-dependent Sobolev spaces $H^p_t H^q_x$, which are crucial for evolutionary PDEs (e.g., Navier–Stokes, Porous Medium).

**Weaknesses:**

- The authors only provide results for the Porous Medium Equation (PME), while the second application, the Compressible Navier–Stokes equation, remains purely theoretical.
- The networks used (single hidden layer PINNs with 2500, 5000, and 7500 nodes) are not common in the PINN literature, where deeper and narrower architectures are typically preferred for complex PDEs.
- The paper suffers from presentation issues. For instance, around line 471, “Figure 1” should likely refer to Table 1, and in Figures 1–2, the blue and orange colors are never explained. Moreover, $N$ is used in the introduction section, but is only formally defined in Section 2, which makes it difficult for readers to follow.
- The paper ends abruptly after the experimental section without a proper Conclusion or Discussion. Moreover, there is no reflection on the limitations of the current analysis.

**Questions:**

- What is the difference between RaNN and ELM proposed by Huang et al. (2006)?
- What is the definition of convergence rate (in line 20)?
- In line 183, what is $\phi(x)$?
- Does equation (14) mean $u_N$ is an unbiased estimator of $u$? or just $\mathbb{E}(u_N) = u$?
- If you mean $\mathbb{E}(u_N) = u$ by "unbiased", does it hold for $\forall N\geq 1$?
- Theorem 1 assumes $t-$like distribution. However, you sampled the weights from the Gaussian distribution in the experiments. Is there a reason for this choice?
- It seems the minimum regularity in Theorem 1 is quite restrictive ($s_1>3/2, s_2>(d+2)/2$). Could you provide the benchmark problems satisfying these conditions?
- Is it possible to check whether the condition in line 345 is satisfied?
- Is it true that one can find such a sequence as in line 349? If so, how?
- The weights in the experiment are sampled from $\mathcal{N}(0, 100I_{d+1})$. What makes the covariance so high compared to the conventional initialization scheme? How did you choose the variance $100I_{d+1}$? How did you initialize the PINNs?
- The choice of PINNs with a single hidden layer and 2500, 5000, 7500 nodes is unrealistic. Could you replace them with typical architectures with a similar number of parameters? For instance, d-32-32-32-1 has about 2000 trainable parameters.
- In Table 1, could you provide the results for RaNN without Fourier features?
- Check the typos (e.g., line 471 Figure 1 -> Table 1).
- Could you provide detailed training settings (e.g., optimizer, learning rate)
- Could you provide more experimental results that support the theoretical claims?
- This paper is under review as a conference paper at ICLR 2026. It seems the paper is written with ICLR 2025 format.

---

> ### Author Response · Authors · 2025-11-22
> **Response (1/2)**
>
> ## Response (1/2)
>
> Firstly, thank you for taking the time to read our manuscript carefully. Your comments have undoubtedly helped us to improve the quality of our work.
> We have uploaded a revised version of the paper in which we address all the points you raised. Below we explain in details the changes made to address all questions and concerns.
>
>
> Before addressing your questions directly, we would like first clarify the contribution of the paper and point out that we have made several major changes to the manuscript.
>
> **Contribution and new experiments**
> We would like to clarify that the main contribution of the paper lies in the novel approximation bounds of Theorem 1: for Randomised Neural Networks (RaNNs) approximating time-dependent Sobolev functions. In particular, our results allow to measure the approximation error on the entire time interval rather than just at a single time point.
>
> The numerical experiments of Section 5 are provided to illustrate the usefulness of the theoretical result. The purpose of these experiments is to support the main message of Theorem 1, which is that RaNNs can efficiently learn complex functions, and in particular functions which are solutions to non-linear PDEs. We have completely reworked Section 5 of the paper to make this point much clearer and have added more extensive experiments.  Our simulations intend to demonstrate that the obtained theoretical guarantees for RaNNs are reflected in practical performance, even in challenging settings such as the PME and compressible Navier-Stokes equation.
>
> 1.  **Additional experiments for PME #1 (Section 5.1.1):** We investigated the effect of the network width $N$ on the error of a RaNN, for the case of the Barenblatt solution to PME in dimensions $d=1,\dots,5$. The plots, which support the theoretical $N^{-1/2}$ rate, have been added to the manuscript.
> 2.  **Additional experiments for PME #2 (Section 5.1.2):** We have added experiments for PME in dimensions $d=4,5$ (see Section 5). The architecture used for PINN (B) has been modified to a deeper, more traditional architecture (four hidden layers with width chosen to match the number of trainable parameters of the RaNN). We have used a hybrid Adams + L-BFGS optimisation set up for all three networks. Full details of optimisation/initialisation procedures have been added to the manuscript.
> 3.  **Experiments for CNS (Section 5.2):** We added experiments which examine the effect of network width $N$ on the error of a RaNN, for solutions to the compressible Navier-Stokes system in $d=1$. The plots, which support the theoretical $N^{-1/2}$ rate, have been added to the manuscript. Extending this experiment to dimensions $d \ge 2$ is currently infeasible because we don't have access to an appropriate 'ground-truth' solution. Indeed, strong solutions to 2D compressible NS equations lack global-regularity guarantees and may blow up in finite time.
> 4.  **Additional experiments for PME #3 (Appendix B.1.3):** We have moved the previous experiments for $d=1,2,3$ to the appendix and added in the case of RaNNs without Fourier feature layer. This can be seen in Appendix B.1.3.
> 5.  **Conclusion:** We have also added a conclusion section to the end of the paper. We now respond to each of your points.
>
> **Weaknesses**
> 1. We now provide additional experiments, both for the PME and the NLS (see above).
> 2. We adjusted the PINN and RaNN architectures as suggested.
> 3. We fixed all the mentioned points and carefully read again the entire manuscript to remove any remaining inaccuracies.
> 4.  Thank you for pointing this out. We have added a conclusion/discussion section to the end of the paper to address this, where we also point out  limitations of the current work that we would consider as interesting starting points for future work.
>
> **Responses to Specific Points**
>
> 1.  In terms of architecture, RaNN is identical with ELM; both are single hidden layer networks in which only the output weights are trainable. ELM is often used to denote a specific algorithm for RaNN training which uses a least-squares pseudo-inverse solution to find the optimal weights. In contrast, as we see it RaNN describes only the architecture; and RaNN can be used as building blocks in other algorithms. For example, for non-linear PDEs a least-squares solution is not applicable directly. However, the terms RaNNs and ELM are sometimes also used interchangeably in the literature.
> 2.  The "convergence rate" is the rate of decay of the error as a function of network width. Specifically, our results show that the error of a neural network $u_N$ approximating some function $u$ will look like $\|u-u_N\| \le C N^{-\alpha}$. In this sense, we say the convergence rate is $\alpha$.
> 3. Thank you for pointing this out, it is a typo. It should be $f$.
>
> **(continued)**

---

> ### Author Response · Authors · 2025-11-22
> **Response (2/2)**
>
> ## Response (2/2)
>
> **Responses to Specific Points (continued)**
>
> 4. Unbiased here just means $\mathbb{E}[u_N] = u$. More in detail, equation (14) is an upper bound for $\mathbb{E}(\|u-u_N\|^2)$, which does not imply unbiasedness alone. However, the estimate $u_N$ does satisfy $\mathbb{E} (u_N) = \tilde{u}$ (the smooth extension of $u$ to full space). By definition this means that $u_N$ is an unbiased estimator of $\tilde{u}$.
> 5.  Yes it is unbiased for all $N \ge 1$, since it is constructed as a Monte-Carlo estimator.
> 6.  In our tests, we found that Gaussian weights were slightly more stable compared to the t-like distribution and produced higher accuracy on average. Since we used Fourier features, the weights correspond to the frequencies of $\sin / \cos$ functions and so higher frequencies help us to represent more complex functions (which is useful for higher $d$ problems). This is why we chose a high variance. This point is clarified in Appendix B.1.2.
> 7.  The Corollaries 1 and 2 in Section 4 give concrete examples of solutions which satisfy our conditions. For example, taking $(\rho_0, u_0) \in H^5(D)$ with $0 < \delta \le \rho_0 < C$ generates a solution $(\rho, u)$ that satisfies the conditions of Theorem 1. Indeed, this is how the high-probability bounds of Corollary 2 are obtained.
> 8.  The condition in line (345) is always satisfied by some $L$. Indeed, any RaNN will be smooth, and therefore it (and its derivatives) will be bounded by some constant $L$. We do not assume anything about $L$. In other words, line 345 is not a condition. We are just labelling an upper bound on the (random quantity) $\|u_N\| + \|\nabla u_N\| +\|\Delta u_N\|$ by $L$.
> 9.  If you pick any $N \in \mathbb{N}$ then Corollary 1 tells us that we can find a neural network $u_N$ so that the PDE residual is bounded by $C/\sqrt{N}$ (where $C$ is everything else in the right hand side of (21)). So if you take $N \to \infty$, you can obtain a sequence of neural networks such that the PDE residual is going to $0$, because of (21).
> 10. Since we used RaNNs with Fourier features, the weights correspond to the frequencies of $\sin / \cos$ functions and so higher frequencies help us to represent more complex functions (which is useful for higher $d$ problems). We found higher variance to lead to a higher accuracy in general. This point is clarified in Appendix B.1.2. For the PINNs we use a standard Xavier (Glorot) initialization.
> 11. We agree that standard PINN implementations typically benefit from depth. In response, we have added **PINN (B)**, a deep network ($d-M-M-M-M-1$) with a parameter count equivalent to the RaNN, to provide a fair comparison against state-of-the-art architectures (see Section 5.1.2 and Appendix B.1.2). However, we maintain the single-layer **PINN (A)** as a baseline to isolate the effect of training hidden weights. Comparing the RaNN to PINN (A) highlights that even with identical shallow architectures, the randomization provides significant optimization benefits.
> 12. Thank you for this point. We followed your suggestion and carried out further experiments. We have added an additional table where we compare PINN (A), PINN (B), RaNN (A) and RaNN (B), where RaNN (B) is an RaNN with no Fourier feature layer. The experiments are collected in Appendix B.1.3 (Table 3).
> 13. Thank you for this. It has been corrected.
> 14. We have added details on the optimiser and learning rate for all experiments now. Thank you very much for pointing out this oversight.
> 15. As discussed above, we have added a number of further experiments (more dimensions for PME, and error rate plots against $N$ for PME and NSE).
> 16. Thank you for pointing this out. We have changed the format to 2026.
>
>
>
> We would like to thank you once again for taking the time to provide your comments. We found them to be very useful in improving the quality of our work.
> We hope that these changes have clarified your questions and concerns.

---

> > ### Comment · Reviewer_TgGQ · 2025-11-27
> >
> > I appreciate the authors’ detailed rebuttal and the considerable effort they have clearly invested in revising the manuscript. The additional experiments, clarifications, and architectural adjustments indicate a sincere attempt to address the reviewers’ concerns, and I thank the authors for engaging constructively with the feedback.
> >
> > After reading both the original submission and the rebuttal, my overall assessment remains largely unchanged. The theoretical contribution, an approximation bound for RaNNs in time-dependent Sobolev spaces, is mathematically sound and represents an extension of prior RaNN/ELM results. However, I am still unsure whether this contribution alone is sufficiently strong or novel to serve as a standalone main result for a top-tier venue like ICLR.
> >
> > A more significant concern is the conceptual mismatch between the theory and the empirical section. Theorem 1 is an expressivity/approximation result, but the experiments focus on PDE-solving performance via PINN-style training. This makes the narrative difficult to interpret. If the goal is to empirically support the theorem, a more natural approach would be to numerically verify the predicted upper bound or to demonstrate expressivity benefits directly. If the goal is to argue that RaNNs can address known optimization issues in PINNs, then evidence should be provided on improved trainability e.g., optimization landscape, convergence behavior, or stability.
> >
> > Instead, the paper compares RaNNs to standard PINNs using minimal or naïve PINN setups. Modern PINN literature already shows that techniques such as adaptive sampling, causal training for time-dependent PDEs, and adaptive loss balancing can significantly improve performance. Without incorporating or comparing against these well-established baselines, the empirical evaluation does not convincingly demonstrate any intrinsic advantage of RaNNs, nor does it reveal a meaningful connection to the theoretical expressivity result.
> >
> > As a result, it remains unclear what the paper ultimately aims to claim:
> > - that RaNNs are theoretically expressive,
> > - that RaNNs are practically more effective for PDE solving, or
> > - that RaNNs could replace PINNs in certain regimes.
> >
> > Currently, the experiments do not strongly support any of these interpretations, leaving the overall message of the paper somewhat ambiguous.
> >
> > I again thank the authors for their extensive rebuttal and revisions. While the work contains interesting theoretical ideas, the gap between the theory and empirical claims, together with the limited experimental rigor relative to modern PINN practice, makes it difficult for me to fully endorse the submission in its present form.

---

> ### Author Response · Authors · 2025-11-27
> **Response**
>
> We thank the reviewer for the reply. We believe that these points are still based on the original submission, rather than the revised pdf that we had uploaded. In the revised manuscript we have put emphasis on the fact that the key contribution of the paper lies in its theoretical results (Theorem 1 and its applications to PINNs for two important classes of non-linear PDEs, see Corollary 1 and 2).
>
> Thereby, our results provide a theoretical foundation to RaNN-based PINNs for non-linear PDEs. While these methods have been extensively used empirically (see the references in lines 60-62), such approximation error guarantees (Corollary 1 and Corollary 2) were not available. This is also the reason why we selected “learning theory” as primary area. In particular, our paper does not make any empirical claims in the revised version.
>
> Moreover, we respectfully disagree with the concern about a *conceptual mismatch*. Our revised version puts emphasis on the fact that the empirical results serve to numerically verify the predicted approximation rate appearing from the theoretical bound (see Figure 1 and Figure 2). We believe that the reviewer may have overlooked these new experimental results in Sections 5.1.1 and 5.2. These experiments serve to **numerically verify the predicted upper bound** (as the reviewer mentions). These results are not obtained via PINN-style training, but by taking the true solution $u$ and training a RaNN to approximate it (which mirrors the result of Theorem 1, where the true solution u is fixed and the RaNN approximation $u_N$ is constructed). The reviewer seems to be only taking into account Section 5.1.2., which serve as a baseline but could be safely moved to the supplementary material. We are happy to do so, if the reviewer thinks that it would help to streamline the presentation.
>
> We would like to thank the reviewer for the feedback and kindly ask to reconsider the assessment, taking into account the above clarification about the papers’ contribution, the fact that all raised concerns about presentation have been incorporated, and the fact that we have addressed all weaknesses mentioned in the original reviewer comments.

---

### Official Review · Reviewer_Mrho · 2025-10-30

**Soundness:** 2
**Presentation:** 2
**Contribution:** 2
**Rating:** 6
**Confidence:** 2

**Summary:**

For random neural networks (RaNNs), the authors derive approximation error bounds for time-dependent Sobolev functions and obtain a dimension-free approximation rate 1/2, inidicating that RaNNs are capable of efficiently approximating solutions to complex nonlinear PDEs. When applied to Physics-Informed Neural Networks (PINNs), the bounds imply that with high probability, the PINN training error converges to 0 with convergence rate free from the curse of dimensionality. Numerical simulations show that RaNN-based PINNs achieve a similar accuracy to classical PINNs, while requiring significantly less training time, supporting the theoretical findings.

**Strengths:**

* (Relevance) A precise theoretical understanding of RaNN approximation capabilities is crucial because the reduced number of trainable parameters of RaNNs compared to fully trainable models and  a simpler training phase with reduced computational cost are sometimes obtained at the expense of lower expressivity. (Lines 72-75)
* (Technical contribution & Novelty) The derived bounds are tailored to time-dependent, non-linear PDE problems, which starkly contrast the paper from previous results, which require strict information on the solution or are applicable only for approximating PDE solutions at a fixed point in time. (Lines 84-91)
* (Technical cotribution) The analysis yields residual-loss bounds for the PME and 1D compressible Navier–Stokes, clarifying when randomized features can train efficiently without the curse of dimensionality.
* (Reproducibility) The code is provided.

**Weaknesses:**

* Discussion on the comparison with other RaNN-based PINNs: \[Nelsen & Sturat 2021; Gonon 2023; Jacquier & Zuric2023；Neufeld+ 2025\] is recommended to highlight the paper's contribution.
* The code is provided, but adding a brief readme file would be helpful.
* Results require smooth solutions ($s_1>3/2, s_2>(d+2)/2$) and boundedness of certain network norms

  for the high-probability residual bounds. Constants like $C\_\\Omega$, $L\_\\psi$, $C\_\\pi$ are intricate and not empirically calibrated. The rate is dimension-independent, but constants could possibly depend on $d$ via extensions/embeddings, to my understanding.
  * Could the authors quantify or upper-bound key constants (e.g., $C\_\\Omega, L\_\\psi, C\_\\pi$) for representative domains and p, q so users can anticipate practical sample complexity?
* Including scaling curves of error/residual and runtime vs. N would be benefricial to empirically support the theoretical convergence rate.

**Questions:**

- Do you foresee extensions to non-smooth or stiff solutions (e.g., shocks), or stochastic PDEs, perhaps via different norms?

---

> ### Author Response · Authors · 2025-11-22
> **Response**
>
> Firstly, we would like to thank you for your careful reading and your useful comments which have helped us to improve the quality of the manuscript. We have uploaded a revised version of the paper in which we address all the points you raised. Below we explain in details the changes made to address all questions and concerns.
>
> Before addressing your questions directly, we would like to point out that we have made several changes to the experiments section:
>
> 1.  **Additional experiments for PME #1 (Section 5.1.1):** We investigated the effect of the network width $N$ on the error of a RaNN, for the case of the Barenblatt solution to PME in dimensions $d=1,...,5$. The plots, which validate the theoretical $N^{-1/2}$ rate, have been added to the manuscript.
> 2.  **Additional experiments for PME #2 (Section 5.1.2):** We have added experiments for PME in dimensions $d=4,5$ (see Section 5). The architecture used for PINN (B) has been modified to a deeper, more traditional architecture (four hidden layers with width chosen to match the number of trainable parameters of the RaNN). We have used a hybrid Adam + L-BFGS optimisation set up for all three networks. The details of optimisation/initialisation procedures have been added to the manuscript.
> 3.  **Experiments for CNS (Section 5.2):** We added experiments which analyse the effect of network width $N$ on the error of a RaNN, for solutions to the compressible Navier-Stokes system in $d=1$. The plots, which support the theoretical $N^{-1/2}$ rate, have been added to the manuscript. Extending this experiment to dimensions $d \ge 2$ is currently infeasible because we don't have access to an appropriate 'ground-truth' solution. Indeed, strong solutions to 2D compressible NS equations lack global-regularity guarantees and may blow up in finite time.
> 4.  **Additional experiments for PME #3 (Appendix B.1.3):** We have moved the previous experiments for $d=1,2,3$ to the appendix and added in the case of RaNNs without Fourier feature layers.
>
> We have also added in a conclusion section to the end of the paper. We now respond to each of your points.
>
> **Answers to specific comments**
>
> 1.  Thank you for this comment. In comparison to existing approximation results for RaNNs (as listed in the last paragraph of Section 1.1), our results allow to handle time-dependent functions in mixed Sobolev spaces, as arise in the context of non-linear PDEs. Solutions often have significantly different behaviour in time versus space (e.g.\ solutions to Navier-Stokes or semi-linear heat equations). Existing results would either require strict information on the solution (e.g.\ finiteness in Barron-ridgelet norms and decay on the Fourier transform of $u$) which is typically not known, or the bounds would be applicable only for approximating PDE solutions at a fixed point in time. We have tried to make this more clear in Sections 1.1 and 1.2.
> 2.  We have added a `README.md` file which contains information on the purpose of each script, and the main parameters which can be controlled.
> 3.  This is true: the constants do depend on the dimension via extensions/embeddings. For general $p,q$ there is no easy upper bound that we are aware of for $C_\Omega$, for example. However, the key emphasis here lies on the convergence rate, which allows to control how the error decreases as the RaNN network size increases (cf.\ also the next comment).
> 4.  We have now added experiments for both the PME and CNS equations that show the effect of $N$ on the relative $L^2$ error. See Figure 1 in Section 5.1.1, Figure 2 in Section 5.2, and Figure 3 in Appendix B.1.2. These plots support the theoretical $N^{-1/2}$ rate suggested by Theorem 1.
> 5. We have no reason to believe that such a result isn't possible. However, our approach to proving Theorem 1 relies heavily on some smoothness of the target solution $u$. We believe that a result for non-smooth/less stiff solutions, if possible, would require a different representation than the method via the ridgelet transform chosen in this paper.
>
>
> We would like to thank you once again for taking the time to provide your comments. We found them to be very useful in improving the quality of our work.
> We hope that these changes have clarified your questions and concerns.

---

> > ### Comment · Reviewer_Mrho · 2025-11-27
> >
> > Please accept my apologies for the delayed response. I deeply appreciate the authors’ detailed reply, including the additional experiments.
> >
> > Some of my concerns are addressed in **Answers to specific comments**, and I would like to add the following remarks:
> >
> > 1. Relevant and related studies are now discussed.
> >
> > 2. A README.md file has been added to the Supplementary Material.
> >
> > 3. Do the authors expect (or can one show under the given assumptions) that these constants can be bounded by a polynomial function of $d$ at most?
> >
> > 4. The paper would benefit from including error bars in figures and tables to improve reproducibility. Moreover, to substantiate the claim that “the theoretical $N^{-1/2}$ is validated,” a more precise analysis is required—for example, taking the logarithm of the decay curve, fitting a linear model, computing the slope, and testing it against $-1/2$. In this sense, the current empirical evaluation could be made more convincing, as noted by Reviewer TgGQ.
> >
> > 5. The discussion could be included in the Appendix as supplementary material.

---

> > > ### Author Response · Authors · 2025-12-03
> > > **Response**
> > >
> > > We would like to thank you once again for your valuable feedback. In response, we have made the following changes.
> > >
> > > 1. We have added log-log plots (Figures 4 and 6) which allow us to further compare the numerical results with the theoretical rate by estimating the slope via a regression. These figures also include error bands which indicate the region within one standard deviation of the mean error.
> > > 2. The original Figures (non-log scale) which investigate error vs network width (Figures 1,2, and 3) now include error bands which indicate the region within one standard deviation of the mean error (across the five runs for each width).
> > > 3. We have added Remark 4 in the appendix (p21) which shows that the constant grows at most polynomially in dimension $d$ whenever the constant $C_\Omega$ also grows polynomially. A sufficient condition for the latter is, e.g., that $p=q=0$ or that $u$ already exists on the full space.
> > >
> > > For the NSE case (Figures 2, 6) the simulation results align with the theoretical results, but for low widths there is more noise than expected. We are currently investigating reasons for this.
> > >
> > > We hope that these changes resolve your remaining concerns formulated in your points 4. and 5.

---

### Official Review · Reviewer_iesZ · 2025-10-30

**Soundness:** 2
**Presentation:** 2
**Contribution:** 2
**Rating:** 4
**Confidence:** 1

**Summary:**

This paper studies RaNN expressivity for learning non-linear PDEs solutions. It derives approximation error bounds for time-dependent Sobolev functions and obtain a dimension-free approximation rate 0.5. The experiments in solving Porous Medium Equations (PME) and Compressible Navier-Stokes Equations shows similar accuracy to the PINN method, while requiring less training time.

**Strengths:**

The proposed method achieves similar performance with less training time and fewer number of parameters compared with PINN.

**Weaknesses:**

This paper lacks ablation test and thus it is difficult to know the robustness and the scalability. For example, compare with PINN with similar number of trainable parameters? If the neural network is scaled to similar size, does the method still require less training time? How does the training time and number of trainable parameters affect the final results? Does it match the theory?

**Questions:**

The proposed method seems to be efficient in model size. Does scaling it further up improve the results?

---

> ### Author Response · Authors · 2025-11-22
> **Response**
>
> Firstly, we would like to thank you very much for your feedback and your useful comments which have certainly helped us to improve the quality of the manuscript. We have uploaded a revised version of the paper in which we address all the points you raised. Below we explain in details the changes made to address all questions and concerns.
>
> Before addressing in detail all the comments, we would like to provide some general comments.
>
> **Contribution and new experiments**
> We would like to clarify that the main contribution of the paper lies in the novel approximation bounds of Theorem 1: for Randomised Neural Networks (RaNNs) approximating time-dependent Sobolev functions. In particular, our results allow to measure the approximation error on the entire time interval rather than just at a single time point.
>
> The numerical experiments of Section 5 are provided to illustrate the usefulness of the theoretical result. The purpose of these experiments is to support the main message of Theorem 1, which is that RaNNs can efficiently learn complex functions, and in particular functions which are solutions to non-linear PDEs. We have completely reworked Section 5 of the paper to make this point much clearer and have added more extensive experiments.  Our simulations intend to demonstrate that the obtained theoretical guarantees for RaNNs are reflected in practical performance, even in challenging settings such as the PME and compressible Navier-Stokes equation.
>
> In particular:
>
> 1. Experiments for the Porous Medium Equation (PME) have been extended to dimensions $d=4$ and $d=5$. The architecture used for PINN (B) has been modified to a deeper, more traditional architecture (four hidden layers with width chosen to match the number of trainable parameters of the RaNN). We have used a hybrid Adams + L-BFGs optimisation for all three networks. Full details of optimisation/initialisation procedures have been added to the manuscript.
> 2. We validate how changing the network width $N$ affects the error of a RaNN approximating PME solution in dimensions $d=1,...,5$. The plots, which validate the theoretical $N^{-1/2}$ rate, have been added to the manuscript.
> 3. We added additional experiments for the compressible Navier-Stokes PDE. We validate how changing the network width $N$ affects the error of a RaNN approximating solutions to the compressible Navier-Stokes system in $d=1$. The plots, which validate the theoretical $N^{-1/2}$ rate, have been added to the manuscript. Extending this experiment to dimensions $d \ge 2$ is currently infeasible because we don't have access to an appropriate 'ground-truth' solution. Indeed, strong solutions to 2D compressible NS equations lack global-regularity guarantees and may blow up in finite time.
> 4. We added experiments with RaNNs without Fourier layers in the Appendix.
>
>
> Moreover, we also added a conclusion section and further discussion on the numerical results.
>
>
> **Responses to specific comments**
>
> Thank you for your comments. As outlined above, we have added more detailed experiments in response to your feedback. The PINN (B) model used in the experiments for the PME (Section 5.2) now uses a similar number of parameters to the RaNN. In this case the RaNN still requires less training time (see Table 1). This is expected since we do not need to use autograd during the backpropagation stage when training the RaNN, since the derivatives of the activation can be hard-coded once and re-used during backpropagation. Moreover, we also added detailed experiments on how changing the network width $N$ affects the error of a RaNN approximating PME solution in dimensions $d=1,...,5$. The plots, which validate the theoretical $N^{-1/2}$ rate, have been added to the manuscript.
>
>
> We hope that these changes have clarified your questions and concerns.

---

> > ### Comment · Reviewer_iesZ · 2025-11-27
> >
> > Thanks for the reply but some of my concerns remain: when you keep increasing the number of parameters in PINN and RaNN, do they always keep a performance gap or they finally end up at the same place? More specifically, another line for PINN in Figure 2 where both curve converges.

---

> > > ### Author Response · Authors · 2025-11-28
> > > **Response**
> > >
> > > We thank the reviewer for the reply. We would like to clarify that our paper does not claim a performance gap between RaNNs and PINNs. Instead, the key contribution of the paper lies in its theoretical results (Theorem 1 and its applications to PINNs for two important classes of non-linear PDEs, see Corollary 1 and 2).
> > >
> > > Thereby, our results provide a theoretical foundation to RaNN-based PINNs for non-linear PDEs. While these methods have been extensively used empirically (see the references in lines 60-62), such approximation error guarantees (Corollary 1 and Corollary 2) were not available. This is also the reason why we selected “learning theory” as primary area. In particular, our paper does not make any empirical claims in the revised version.
> > >
> > > In particular, Figure 2 is not concerned with PINNs vs. RaNNs, but instead we compare the PDE solution to its RaNN approximation to verify the validity of the convergence rate obtained in Theorem 1.
> > >
> > > We hope that this clarifies the reviewer’s questions and concerns.

---

### Official Review · Reviewer_2FSb · 2025-10-31

**Soundness:** 3
**Presentation:** 2
**Contribution:** 2
**Rating:** 2
**Confidence:** 4

**Summary:**

The authors study the expressivity of Random Neural Networks (RaNNs), deriving RaNN approximation error bounds for time-dependent Sobolev functions. They show that RaNNs can approximate solutions to complex nonlinear PDEs with a dimension-free approximation rate (the generic Monte Carlo Rate, $N^{-1/2}$, with N being the number of neurons). The integral representation of the solution using a Ridgelet transform is used to derive the error bounds. The authors improve results in Theorem 3.9 from De Ryck et al. (2025) and extend results to time-dependent norms. The authors also demonstrate, in a computational experiment, that RaNNs are faster to train for similar accuracy compared to PINNs on Porous Medium Flow (PME) example.

**Strengths:**

1) The derivation of approximation error bounds including the dimension-free approximation rate for time-dependent Sobolev functions is a significant theoretical contribution that extends previous works. For random feature methods, the rate $N^{-1/2}$ is very common, but the special type of functions addressed here makes the result relevant for the PDE setting.
2) Considering time dependence in the approximation is important, as most related theory concerns (linear) static PDE.
3) The main contribution of the paper, the proof of theorem 1, is featured prominently and explained well. Using ridgelet transforms is an interesting and useful idea in this field.

**Weaknesses:**

1) It is not clear how time-dependence is special/important to the theoretical results. I might have missed it when reading the proofs, but the dependence on time is not different from any other spatial dimension (also when looking at the model in eq. 1, time is treated like this). The same can be said for "nonlinearity" of the PDEs involved, as the theoretical results are related to general Sobolev functions, not the specific PDEs that have them as solution.

2) The write-up seems partially incomplete: (1) Conclusion/discssion section with future work is missing, (2) limitations are not addressed explicitly.

3) A single benchmark is not enough to demonstrate the theoretical results, especially since many computational results for RaNNs exist already (see list in "questions"). The experimental section is quite weak compared to the typical benchmarking standards of the ICLR community. More examples (e.g., d > 3 for PME or compressible Navier-Stokes equations) would strengthen the claims; also, as the main result is the independence of the approximation rate w.r.t. dimension, a computational experiment with increasing input dimension (e.g. from 2-10) that demonstrates this independence would have been useful.

**Questions:**

1) In Figures 1 and 2, is the true solution given by the blue or orange line?

2) Did you try Porous Medium Equation for dimensions beyond three? What happens for higher dimension beyond 3?

3) Did you try experiments with compressible Navier-Stokes equations? The current setup sets p=0 and mu(rho)=rho, which is extremely simplistic. Solving NS in the turbulent regime presents challenges beyond the curse of dimensionality, more related to the issue of spectral bias of networks. Is the spectral bias also something that random networks can resolve better than the ones trained with gradient descent?

4) The empirical details are not fully clear. Which optimizer was used to compare the results with PINNs? If a first-order optimizer like Adam is used, a comparison with any second-order optimizer (generally considered state-of-the-art for PINNs) would be helpful. E.g., L-BFGS, Adam + L-BFGS, SSBryoden.

5) RaNN-based PINN performance, including high-dimensional examples, has been demonstrated  extensively in computational experiemnts, which should be cited and discussed. Examples:
 - Sun, Jingbo, Suchuan Dong, and Fei Wang. 2024. “Local Randomized Neural Networks with Discontinuous Galerkin Methods for Partial Differential Equations.” Journal of Computational and Applied Mathematics 445 (August): 115830. https://doi.org/10.1016/j.cam.2024.115830.
 - Datar, Chinmay, Taniya Kapoor, Abhishek Chandra, et al. 2024. “Solving Partial Differential Equations with Sampled Neural Networks.” arXiv:2405.20836. Preprint, arXiv, May 31. http://arxiv.org/abs/2405.20836.
 - Nelsen, Nicholas H., and Andrew M. Stuart. 2021. “The Random Feature Model for Input-Output Maps between Banach Spaces.” SIAM Journal on Scientific Computing 43 (5): A3212–43. https://doi.org/10.1137/20M133957X.
 - Chen, Jingrun, Xurong Chi, Weinan E, and Zhouwang Yang. 2022. “Bridging Traditional and Machine Learning-Based Algorithms for Solving PDEs: The Random Feature Method.” arXiv:2207.13380. Preprint, arXiv, July 27. http://arxiv.org/abs/2207.13380.

6) For general random feature networks in the supervised learning setting, tighter bounds were proven already (albeit not in the Sobolev case), see here:
Rudi, Alessandro, and Lorenzo Rosasco. 2021. “Generalization Properties of Learning with Random Features.” arXiv:1602.04474. Preprint, arXiv, April 15. http://arxiv.org/abs/1602.04474.

Would it be possible to tighten the bounds proven in the present work using their methods?

---

> ### Author Response · Authors · 2025-11-22
> **Response (1/2)**
>
> ## Response (1/2)
>
> Firstly, we would like to thank you very much for your careful reading of the paper and your useful comments which have certainly helped us to improve the quality of the manuscript. We have uploaded a revised version of the paper in which we address all the points you raised. Below we explain in details the changes made to address all questions and concerns.
>
> Before addressing in detail all the comments, we would like to provide some general comments.
>
> **Contribution and new experiments**
> We would like to clarify that the main contribution of the paper lies in the novel approximation bounds of Theorem 1: for Randomised Neural Networks (RaNNs) approximating time-dependent Sobolev functions. In particular, our results allow to measure the approximation error on the entire time interval rather than just at a single time point.
>
> The numerical experiments of Section 5 are provided to illustrate the usefulness of the theoretical result. The purpose of these experiments is to support the main message of Theorem 1, which is that RaNNs can efficiently learn complex functions, and in particular functions which are solutions to non-linear PDEs. We have completely reworked Section 5 of the paper to make this point much clearer and have added more extensive experiments.  Our simulations intend to demonstrate that the obtained theoretical guarantees for RaNNs are reflected in practical performance, even in challenging settings such as the PME and compressible Navier-Stokes equation.
>
> In particular:
>
> 1. **Additional experiments for PME #1 (Section 5.1.2)** Experiments for the Porous Medium Equation (PME) have been extended to dimensions $d=4$ and $d=5$. The architecture used for PINN (B) has been modified to a deeper, more traditional architecture (four hidden layers with width chosen to match the number of trainable parameters of the RaNN). We have used a hybrid Adams + L-BFGs optimisation for all three networks. Full details of optimisation/initialisation procedures have been added to the manuscript.
> 2. **Additional experiments for PME #2 (Section 5.1.1)** We validate how changing the network width $N$ affects the error of a RaNN approximating PME solution in dimensions $d=1,...,5$. The plots, which validate the theoretical $N^{-1/2}$ rate, have been added to the manuscript.
> 3. **Experiments for CNS (Section 5.2):** We added additional experiments for the compressible Navier-Stokes PDE. We validate how changing the network width $N$ affects the error of a RaNN approximating solutions to the compressible Navier-Stokes system in $d=1$. The plots, which validate the theoretical $N^{-1/2}$ rate, have been added to the manuscript. Extending this experiment to dimensions $d \ge 2$ is currently infeasible because we don't have access to an appropriate 'ground-truth' solution. Indeed, strong solutions to 2D compressible NS equations lack global-regularity guarantees and may blow up in finite time.
> 4. **Additional experiments for PME #3 (Appendix B.1.3):** We added experiments with RaNNs without Fourier layers in the Appendix.
>
> Moreover, we also added a conclusion section and further discussion on the numerical results.
>
> **Responses to specific comments**
> We can now respond to each of the weaknesses which you have raised:
>
> 1. **Weakness 1:** We agree that from a purely functional perspective, the time variable enters the argument in the same way as any other coordinate, as reflected in the structure of the model in eq (1). However, many PDEs have different regularity properties in the space and time dimensions, hence the time dimension needs to be singled out. In particular, our approximation bounds are directly relevant for time-dependent PDEs where regularity is expressed in mixed Sobolev norms (i.e. $H^p_t H^q_x$; see for example the regularity of the solution $(\rho,u)$ to CNS in eq (25)). Our bounds explicitly cover this setting, whereas previous results address only static functions and don't provide rates for mixed norms which are crucial for non-linear PDEs.
>
> Similarly, when we refer to non-linearity of PDEs, we don't mean that the theorem depends on the structure of a PDE; indeed, Theorem 1 only considers an arbitrary Sobolev function. Rather, the point is that Theorem 1 covers general time-dependent Sobolev functions which include solutions as non-linear PDEs as a subclass. In other words, the theorem is not PDE-specific but its applicability to solutions to non-linear PDEs is the main motivation for developing such bounds. Applying the result to non-linear PDEs is then the purpose of Section 4, where we directly apply Theorem 1 to two non-linear PDEs and demonstrate high-probability bounds. These proofs serve as a general strategy, which can be repeated for other PDEs of course. Note that previous results only address errors at specific time-points and would thus not be applicable to obtain results as those obtained in Section 4.
>
> **(continued)**

---

> > ### Author Response · Authors · 2025-11-22
> > **Response (2/2)**
> >
> > ## Response (2/2)
> > **Responses to specific comments (continued)**
> >
> > 2. **Weakness 2:** Thank you for pointing this out. We have added a conclusion/discussion section to the end of the paper to address this, where we also point out limitations of the current work that we would consider as interesting starting points for future work.
> >
> > 3. **Weakness 3:** As mentioned above, we have now extended the experiments in Section 5 significantly. The rate $N^{-1/2}$ in particular is supported by the experiments for the PM/CNS equations.
> >
> > **Questions**
> > We now answer each of your questions.
> >
> > 1. The true solution is in orange. We have added this into the caption.
> > 2. We have now added experiments for dimensions $4$ and $5$.
> > 3. The set-up of $p=0$ and $\mu = cst$ is only for simplicity of the proofs, but it can be extended to a more general case where $p, \mu$ are convex (typical assumption for CNS). We have added an experiment which compares the effect of network width $N$ with the RaNN error for the compressible NS equations in $d=1$. We have used the travelling wave solutions as a benchmark. We then use a RaNN to solve the PDE for the travelling wave solution for different values of $N$. The resulting plot can be seen in Section 5. Finally, we now also discuss spectral bias as an important direction of future research in the conclusion.
> > 4. We have added full details on the optimiser / initialisation to the paper. For all three networks, we now use a hybrid Adam + L-BFGs setup, where Adam is used for 4750 epochs and then L-BFGS for the remaining 250 epochs. This split is chosen to keep simulations computationally reasonable, and also since in our tests we do not find major benefit in increasing the L-BFGs epochs past $250$.
> > 5. Thank you for pointing this out. We have now extended the discussion in the introduction and, in particular, referenced these papers.
> > 6. Thank you for pointing this out. The seminal paper by Rudi and Rosasco could indeed provide useful when analysing also the sampling error for RaNN-based learning based on samples. However, in our case we are focus on the approximation error; for this error component the analysis in Rudi and Rosasco would not be applicable.
> >
> > We would like to thank you once again for taking the time to provide your comments. We found them to be very useful in improving the quality of our work.
> > We hope that these changes have clarified your questions and concerns.

---

### Author Response · Authors · 2025-12-03
**Summary of Rebuttal Phase**

# Summary of Rebuttal Phase

We were very sorry to hear about the leak and the disruption of the review process. Unfortunately, our reviewers did not have a chance to reply to our clarifications and revisions before the disruption. To aid the review process, we thus provide a short summary of the main concerns raised by reviewers and our responses.

The main concern raised was that that *experiments were not sufficiently detailed*. We would like to emphasize that the main contribution of our work is the theoretical result of **Theorem 1**, which derives novel approximation bounds for Randomised Neural Networks (RaNNs) approximating time-dependent Sobolev functions. Nonetheless, we made significant changes to our experiments, which we now discuss in detail.

---

## Weakness 1: Experiments

Each of the reviewers asked to see more detailed experiments. More specifically, three main points were raised in all reviews. Below we list each of these points and explain what we did to address it.


1. **More experiments**
We carried out numerous additional experiments to help validate the results of Theorem 1:
   -> We added new experiments (Figures 1,2,3,4,6) to validate the rate of convergence asserted by our theoretical result.

   -> We extended previous experiments to higher dimensions (\(d=4,5\); see Table 2 in Appendix B.1.2).

   -> We added experiments for the compressible Navier-Stokes equations (Section 5.2) to help support the rate of convergence asserted by Theorem 1.

   -> We included additional experiments for a fourth type of network (RaNN with no Fourier Features; see Table 3 in Appendix B.1.4).

3. **Experimental details**
A few reviewers (e.g., 2FSb, TgcQ) pointed out that certain experimental details (e.g., choice of optimiser) were missing.
   -> These details have now been added.

4. **Relevance of experiments**
   - Reviewers Mrho and TgcQ asked for experiments that directly validate the rate of convergence asserted by Theorem 1.
   -> These experiments have been added (see Figures 1 and 2 in Sections 5.1.1 / 5.2).
   - TgcQ also suggested using a more traditional (deeper) architecture for comparison with PINNs in Section 5.1.2.
   -> We followed this suggestion by changing the architecture of PINN (B) to have four hidden layers, with width chosen so that the total number of trainable parameters is close to that of the RaNN.

---

## Weakness 2: Lack of Discussion of Results

Some reviewers pointed out that there was no conclusion/discussion section.
-> We fixed this by adding a **'Conclusions' section** on p.10.

---

## Weakness 3: Presentation Issues / References

- Originally, there were minor presentation issues (e.g., 'Figure' was used to refer to a 'Table').
-> These have all been fixed.
- Some reviewers (e.g., Mrho, 2FSb) asked us to include additional references or discuss certain references further.
-> To address this, we added the missing references and explained further the connection between our results and the highlighted references.

---

### Meta-Review · Area_Chair_4sz9 · 2026-01-09

**Summary:**

Reviewers generally note:
1. Analysis of time-depend PDEs is important and the work achieves rate improvements compared to previous work.
2. Experimental section does not contain enough experiments to justify theoretical results or demonstrate usefulness of this framework.
3. The paper lack baseline comparisons to PINNs of equivalent size or traditional numerical methods for high dimensional PDEs e.g. Monte Carlo methods based on Feynman-Kac formula.
4. Results claiming dimension-free approximation rates are overstated since the required smoothness depends on the dimension.
5. The work has presentation issues such lack of conclusion.

**Reviewer Concerns:**

The authors have added addition results for the compressible Navier-Stokes example and more dimensions for the heat equations. They have also added a PINN baseline and addressed issues of presentation. While I think the paper makes strong theoretical contributions to the theory of random neural networks, I do not believe it is ready for publication or that ICLR is the right venue. Numerical experiments need to be fleshed out a lot more and dependence on dimension needs to be studied much more carefully (adding d=4,5 is far from enough). Furthermore, why are these methods useful? Are they able to solve certain problems other cannot? Can they do it faster and more accurately? Looking at the PINN comparison tables, answers to these questions are unclear; cost-accuracy curves are needed. Furthermore, theoretical results should not be overstated. Claims of "dimension-free" approximation to general Sobolev functions are misleading and should not be made. Sobolev spaces have sharp estimates on both their continuous, non-linear, n-widths and their metric entropy, incurring the curse of dimensionality. Therefore there are no reasonable (ones which do not violate information-theoretic bounds) approximation methods which are dimension-free. The results of Theorem 1 are not dimension-free as the required regularity depends on the dimension of the problem. This fact does not make Theorem 1 uninteresting, contrary, it is a strong result but it should be described for what it actually is.

**Reviewer Scores:**

Reviewer scores may have slightly increased but not enough to merit publication.

---

### Decision · Program_Chairs · 2026-01-26

Reject